# scReadSim: a single-cell RNA-seq and ATAC-seq read simulator

Guanao Yan[1], Dongyuan Song [2] & Jingyi Jessica Li [1,2,3,4,5,6] ✉

Benchmarking single-cell RNA-seq (scRNA-seq) and single-cell Assay for Transposase-Accessible Chromatin using sequencing (scATAC-seq) computational tools demands simulators to generate realistic sequencing reads. However, none of the few read simulators aim to mimic real data. To fill this gap, we introduce scReadSim, a single-cell RNA-seq and ATAC-seq read simulator that allows user-specified ground truths and generates synthetic sequencing reads (in a FASTQ or BAM file) by mimicking real data. At both read-sequence and read-count levels, scReadSim mimics real scRNA-seq and scATAC-seq data. Moreover, scReadSim provides ground truths, including unique molecular identifier (UMI) counts for scRNA-seq and open chromatin regions for scATAC-seq. In particular, scReadSim allows users to design cell-type-specific ground-truth open chromatin regions for scATAC-seq data generation. In benchmark applications of scReadSim, we show that UMI-tools achieves the top accuracy in scRNA-seq UMI deduplication, and HMMRATAC and MACS3 achieve the top performance in scATAC-seq peak calling.

The development of single-cell sequencing technologies has enabled the characterization of genomic, epigenomic, and transcriptomic features of individual cells[1]. More than a thousand computational tools have been developed to analyze single-cell sequencing data[2], necessitating third-party benchmarking of computational tools. Realistic simulators are essential for fair benchmarking, because they can generate synthetic data that mimic real data and contain ground truths for evaluating computational tools.

Although many simulators have been developed for the two most popular single-cell sequencing technologies—single-cell RNA sequencing (scRNA-seq) and single-cell Assay for Transposase-Accessible Chromatin using sequencing (scATAC-seq), most simulators do not generate sequencing reads. Instead, they only simulate read counts in genes[3-7] or genomic regions[8], and we refer to them as "count simulators" (see ref. 9 for a comprehensive review of scRNA-seq count simulators). As such, these count simulators cannot be used to benchmark read-level bioinformatics tools that process reads stored in a FASTQ or BAM file (Supplementary Fig. 1). Exemplary read-level tools include the unique molecular identifier (UMI) deduplication tools for

scRNA-seq data[10-13] and the peak-calling tools for scATAC-seq data[14-17]. Benchmarking these tools requires simulators that can generate synthetic sequencing reads, and we refer to such simulators as "read simulators."

For scRNA-seq, minnow[18] and Dropify[19] are the only two read simulators to our knowledge. Only minnow has a publicly available software package, which inputs a UMI count matrix and a gene annotation file but does not learn from real scRNA-seq reads. Moreover, minnow generates scRNA-seq reads only from annotated genes, so it cannot resemble real scRNA-seq data in intergenic regions that might be transcribed. For scATAC-seq, SCAN-ATAC-Sim[20] is the only read simulator to our knowledge. However, similar to minnow, SCAN-ATAC-Sim does not learn from real scATAC-seq reads. Instead, it generates scATAC-seq reads from bulk ATAC-seq reads under simplistic assumptions: treating genomic regions (peaks called from bulk ATAC-seq reads) as independent, with a maximum of 2 reads allowed in each genomic region per cell, irrespective of the region's length. Moreover, SCAN-ATAC-Sim does not provide synthetic read sequences with quality scores in a FASTQ or BAM file. Rather, it generates a BED file

[1]Department of Statistics, University of California, Los Angeles, CA 90095-1554, USA. [2]Bioinformatics Interdepartmental Ph.D. Program, University of California, Los Angeles, CA 90095-7246, USA. [3]Department of Human Genetics, University of California, Los Angeles, CA 90095-7088, USA. [4]Department of Computational Medicine, University of California, Los Angeles, CA 90095-1766, USA. [5]Department of Biostatistics, University of California, Los Angeles, CA 90095-1772, USA. [6]Radcliffe Institute for Advanced Study, Harvard University, Cambridge, MA 02138, USA. ✉e-mail: jli@stat.ucla.edu

that only contains read start and end positions, thereby limiting direct benchmarking of read-level tools that typically require input of a FASTQ or BAM file. Additionally, we found that installing the minnow and SCAN-ATAC-Sim software packages in C + + was non-trivial due to the specific operating system and compiler requirements.

Motivated by the limitations of existing read simulators (a detailed comparison of scReadSim with the existing read simulators is in Supplementary Table 1), we developed scReadSim as a scRNA-seq and scATAC-seq read simulator that generates realistic synthetic data by mimicking real data. scReadSim inputs a scRNA-seq or scATAC-seq BAM file and outputs synthetic reads in a FASTQ or BAM file (Fig. 1a, b). We verified that scReadSim generates synthetic scRNA-seq and scATAC-seq data that resemble real data at both the read-sequence and read-count levels. Moreover, scReadSim provides ground truths, including unique molecular identifier (UMI) counts for scRNA-seq and open chromatin regions for scATAC-seq. In particular, scReadSim enables users to specify cell-type-specific open chromatin regions for scATAC-seq read generation, and it also allows users to vary the cell number and sequencing depth of synthetic data. In this work, we use two exemplary applications to highlight scReadSim's utility as a benchmarking tool. First, for scRNA-seq, scReadSim mimics real data by first generating realistic UMI counts and then simulating reads.

Hence, the synthetic UMI count matrix—an intermediate output of scReadSim—serves as the ground truth for benchmarking scRNA-seq UMI deduplication tools such as UMI-tools[10], cellranger[11], STARsolo[12], and Alevin[13], which all process reads (in a FASTQ file) into a UMI count matrix. Our benchmarking results indicate that UMI-tools achieves the best accuracy, while Alevin and STARsolo are most computationally efficient. Second, for scATAC-seq, scReadSim provides ground-truth peaks and non-peaks by learning from user-specified trustworthy peaks and non-peaks or by calling trustworthy peaks and non-peaks from real data. The ground-truth peaks provided by scReadSim allow benchmarking peak-calling tools such as MACS3[14], HOMER[15], HMMRATAC[16], and SEACR[17]. Our benchmarking results show that HMMRATAC and MACS3 are the top performers.

## Results

### scReadSim generates realistic synthetic scRNA-seq reads

To evaluate scReadSim's ability to generate realistic scRNA-seq reads, we trained scReadSim on the RNA-seq modality of a mouse 10x single-cell Multiome dataset[21] and compared scReadSim's synthetic data with the real data at two levels: UMI counts and read sequences. We also compared scReadSim with minnow, the only scRNA-seq read simulator with a publicly available software package.

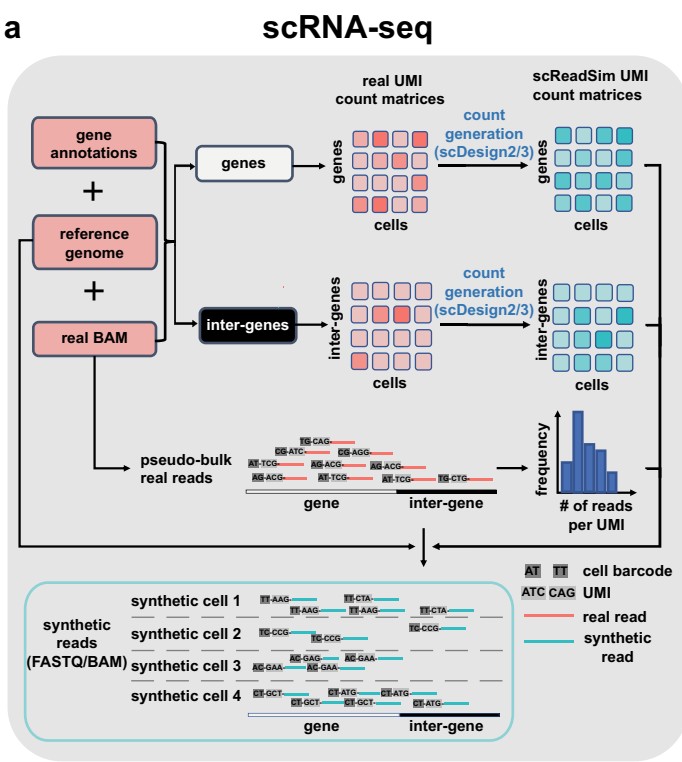

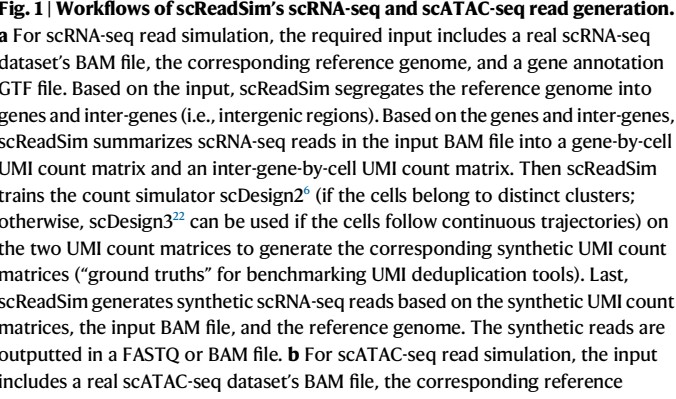

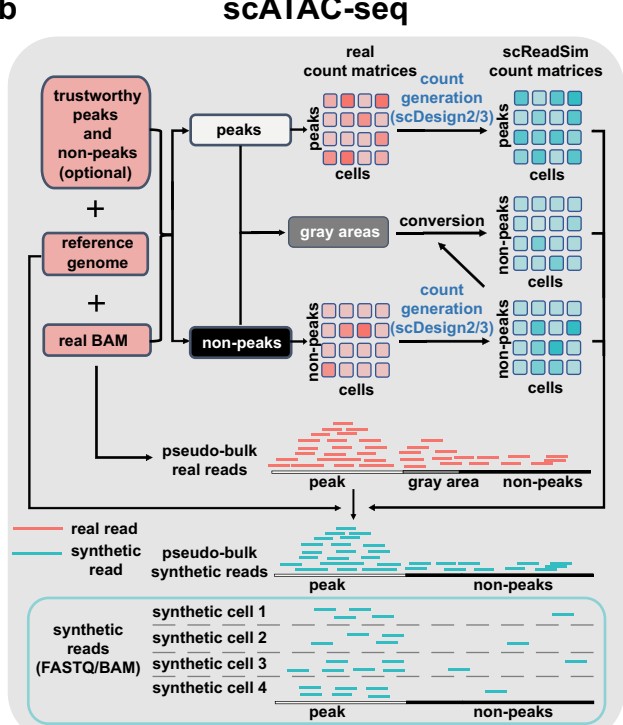

**Fig. 1 | Workflows of scReadSim's scRNA-seq and scATAC-seq read generation. a** For scRNA-seq read simulation, the required input includes a real scRNA-seq dataset's BAM file, the corresponding reference genome, and a gene annotation GTF file. Based on the input, scReadSim segregates the reference genome into genes and inter-genes (i.e., intergenic regions). Based on the genes and inter-genes, scReadSim summarizes scRNA-seq reads in the input BAM file into a gene-by-cell UMI count matrix and an inter-gene-by-cell UMI count matrix. Then scReadSim trains the count simulator scDesign2[6] (if the cells belong to distinct clusters; otherwise, scDesign3[22] can be used if the cells follow continuous trajectories) on the two UMI count matrices to generate the corresponding synthetic UMI count matrices ("ground truths" for benchmarking UMI deduplication tools). Last, scReadSim generates synthetic scRNA-seq reads based on the synthetic UMI count matrices, the input BAM file, and the reference genome. The synthetic reads are outputted in a FASTQ or BAM file. **b** For scATAC-seq read simulation, the input includes a real scATAC-seq dataset's BAM file, the corresponding reference genome, and optionally, users' trustworthy peaks and non-peaks in the input BAM file; if users do not input trustworthy peaks and non-peaks, scReadSim provides two options; see the subsection "scReadSim for scATAC-seq" for detail. Based on the trustworthy peaks and non-peaks, scReadSim defines the complementary genomic regions as gray areas and summarizes scATAC-seq reads in the input BAM file into a peak-by-cell count matrix and a non-peak-by-cell count matrix. Next, scReadSim trains the count simulator scDesign2[6] (or scDesign3[22]) on the two count matrices to generate the corresponding synthetic count matrices for the peaks and non-peaks. Further, scReadSim converts the gray areas into non-peaks (so that the peaks can be regarded as "ground-truth peaks") and constructs a synthetic count matrix based on the gray areas' lengths and the already-generated synthetic non-peak-by-cell count matrix. Last, scReadSim generates synthetic reads based on the three synthetic count matrices, the input BAM file, and the reference genome. The synthetic reads are outputted in a FASTQ or BAM file.

At the UMI-count level, scReadSim uses the realistic count simulator scDesign2[6] or scDesign3[22] to generate UMI counts (Fig. 1a; "Methods"), and both scDesign2 and scDesign3 can generate more realistic UMI counts under more comprehensive settings compared with other count simulators. Hence, we only conducted a confirmation study by demonstrating that scReadSim's synthetic UMI counts resemble real UMI counts in three aspects: (1) the distributions of six summary statistics, including four gene-level statistics (mean, variance, coefficient of variance, and zero proportion) and two cell-level statistics (zero proportion and cell library size) (Supplementary Fig. 2a); (2) correlations among the top expressed genes (Supplementary Fig. 2b); and (3) cells' UMAP embeddings (Supplementary Fig. 2c).

At the read-sequence level, we evaluated the resemblance of scReadSim's synthetic reads to real reads in three aspects. First, scReadSim's synthetic reads preserve the k-mer spectra, which describe the distribution of k-mers' frequencies (i.e., numbers of occurrences) in reads. Supplementary Fig. 3a shows similar k-mer spectra in synthetic reads and real reads. Second, scReadSim can introduce substitution errors to imitate those observed in real reads. Supplementary Fig. 3b shows that scReadSim's synthetic reads mimic real reads in terms of the substitution error rate per base. Third, after read alignment, the pseudo-bulk read coverage of scReadSim's synthetic reads mimics that of real reads (Spearman correlation = 0.930 in Fig. 2a; Supplementary Fig. 3c). In contrast, the pseudo-bulk read coverage of minnow's synthetic reads does not mimic real data (Spearman correlation = 0.065 in Fig. 2a), an expected result as minnow does not learn from real scRNA-seq reads.

## scReadSim generates realistic synthetic scATAC-seq reads

For scATAC-seq, scReadSim provides ground-truth peaks and non-peaks for benchmarking purposes, by learning from user-specified or scReadSim-identified (under stringent criteria) peaks and non-peaks that are deemed trustworthy in real data (Fig. 1b; "Methods"). First, scReadSim sets trustworthy peaks and non-peaks as ground-truth peaks and non-peaks, respectively. Then regarding the gray areas (i.e., regions complementary to trustworthy peaks and non-peaks), whose read coverages are between those of trustworthy peaks and trustworthy non-peaks in real data, scReadSim converts the gray areas to ground-truth non-peaks ("Methods").

We verified that scReadSim generates realistic synthetic scATAC-seq reads at two levels: read counts and read sequences, by mimicking two real datasets: the ATAC-seq modality of a mouse 10x single-cell Multiome dataset[21] and a sci-ATAC-seq dataset[23]. We also compared scReadSim with SCAN-ATAC-Sim, the only scATAC-seq read simulator available.

At the read-count level, we first confirmed that scReadSim's synthetic peak-by-cell and non-peak-by-cell count matrices mimic the real count matrices in four aspects: (1) the distributions of six summary statistics, including four peak-level statistics (mean, variance, coefficient of variance, and zero proportion) and two cell-level statistics (zero proportion and cell library size) (Supplementary Figs. 4a and 5a); (2) the distributions of normalized read counts (in Reads Per Kilobase Million (RPKM)) of peaks and non-peaks (Supplementary Figs. 4b and 5b); (3) correlations among the top open peaks (Supplementary Figs. 4c and 5c); and (4) cells' UMAP embeddings (Supplementary Figs. 4d and 5d). Second, we found that scReadSim outperforms SCAN-ATAC-Sim in mimicking the sci-ATAC-seq data in both the distributions of the six summary statistics (Supplementary Fig. 6a) and the cells' UMAP embeddings (Fig. 2b and Supplementary Fig. 6b), an expected result as SCAN-ATAC-Sim does not learn from real scATAC-seq data.

At the read-sequence level, we evaluated the resemblance of scReadSim's synthetic scATAC-seq reads to real reads in five aspects.

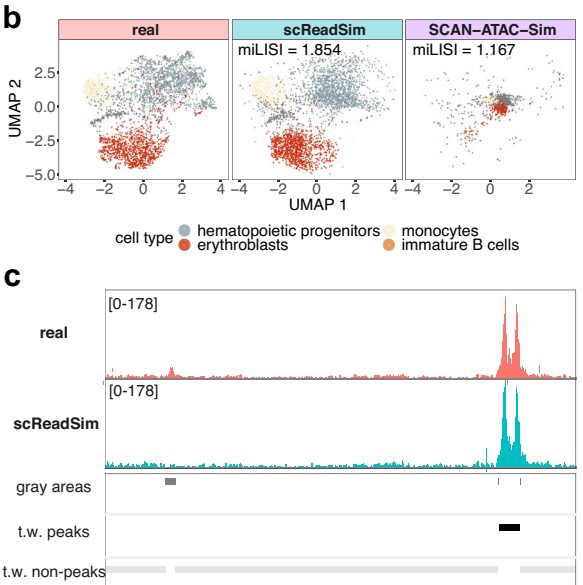

**Fig. 2 | scReadSim outperforms existing scRNA-seq and scATAC-seq read simulators. a** scReadSim outperforms the existing scRNA-seq read simulator minnow in preserving the read coverage in a mouse 10x single-cell Multiome dataset (the RNA-seq modality only)[21]. With chromosome 1 of the reference genome divided into consecutive, non-overlapping 1000-bp windows, the x-axis represents the windows' positions along chromosome 1, and the y-axis indicates the number of the reads overlapping each window in each pseudo-bulk sample (real or synthetic, with all cells pooled). The track height is set to 6,595,507 for all three tracks. The Spearman correlation (Cor.) and Pearson Cor. measure the similarity of read coverage between real and synthetic data across the windows. Inset: a closer view of the region chr1:86,425,858–86,443,378 using the IGV genome browser[46]. Each coverage track displays the depth of the reads covered at each locus and indicates its track height 136 or 438 at the left corner. **b** In UMAP visualizations, scReadSim outperforms the existing scATAC-seq read simulator SCAN-ATAC-Sim in mimicking real cells in the mouse 10x single-cell Multiome dataset (the ATAC-seq modality only)[21]. The miLISI measures the similarity of real and synthetic cells in the UMAP space: the miLISI value ranges between 1 and 2, with 2 indicating a perfect mixing of real and synthetic cells. **c** scReadSim converts gray areas in real data to ground-truth non-peaks in synthetic data, maintaining trustworthy (t.w.) peaks and non-peaks in real data as ground-truth peaks and non-peaks, respectively, in synthetic data.

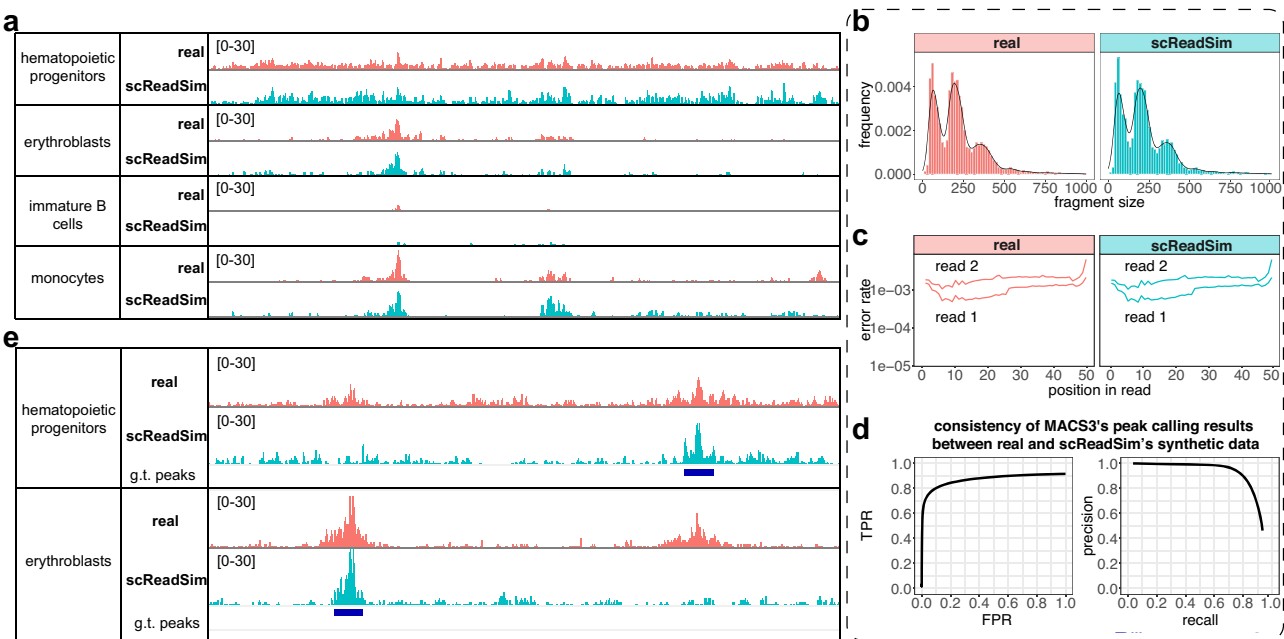

**Fig. 3 | scReadSim generates realistic scATAC-seq sequencing reads with ground-truth peaks.** scReadSim's synthetic scATAC-seq data mimic the real sci-ATAC-seq dataset[23] in terms of the cell-type-specific read coverage (**a**), the fragment-size distribution (**b**), the substitution error rate per base within a read (**c**), and the peaks called by MACS3 at the pseudo-bulk level (**d**; "Methods"). **e** scReadSim enables user-designed, cell-type-specific ground-truth (g.t.) peaks for scATAC-seq read generation.

First, after read alignment, scReadSim's synthetic read coverage mimics the real read coverage in the trustworthy peaks and non-peaks at the pseudo-bulk level; moreover, scReadSim's synthetic read coverage in the ground-truth non-peaks converted from gray areas mimics the real read coverage in the nearby trustworthy non-peaks (Fig. 2c and Supplementary Fig. 7a). We further verified that scReadSim preserves the cell-type-specific read coverage in real data (Fig. 3a). Second, scReadSim's synthetic reads mimic real reads in terms of k-mer spectra (Supplementary Figs. 7b and 8a). Third, scReadSim's synthetic reads preserve the fragment size distribution of real reads (Fig. 3b and Supplementary Fig. 8b); it is known that the fragment size distribution is resulted from the pattern of chromatin accessibility and thus meaningful[24]. Fourth, scReadSim's synthetic reads maintain the substitution error rate per base in real reads (Fig. 3c and Supplementary Fig. 8c). Fifth, MACS3, a popular peak calling tool, obtains consistent peak calling results from scReadSim's synthetic reads and real reads (Fig. 3d, Supplementary Figs. 7c and 8d–e; a reasonable result is that the consistency is higher for the ATAC-seq modality in the mouse 10x single-cell Multiome dataset, which is less sparse than the sci-ATAC-seq dataset).

Beyond mimicking real data, scReadSim also allows users to design ground-truth peaks and non-peaks for synthetic scATAC-seq data generation at the cell-type level (Fig. 3e) or even with an arbitrary design (Supplementary Fig. 9). We show that scReadSim's synthetic scATAC-seq data with user-designed ground-truth peaks and non-peaks preserve the real read-sequence characteristics and mimic the trustworthy peaks and non-peaks in real data (Supplementary Fig. 10). In summary, scReadSim can provide reliable ground truths for benchmarking computational tools that process scATAC-seq reads.

**scReadSim allows varying cell numbers and sequencing depths**
scReadSim flexibly allows users to vary cell numbers and sequencing depths to generate synthetic data. For UMI-based scRNA-seq data, scReadSim defines the baseline sequencing depth as the number of UMIs in the input real data; for non-UMI-based scRNA-seq data and scATAC-seq data, scReadSim defines the baseline sequencing depth as the number of sequencing reads (with duplicated reads resulted from polymerase chain reaction (PCR) removed) in the input real data.

For demonstration purposes, we deployed scReadSim to the RNA-seq modality of a mouse 10x single-cell Multiome dataset[21] and varied the cell number or the sequencing depth (Supplementary Fig. 11). Supplementary Fig. 11a shows the synthetic datasets scReadSim generated with a fixed sequencing depth but increasing cell numbers 1219 (0.25×), 2438 (0.5×), 4877 (1×, the original cell number), 9754 (2×), and 19,508 (4×). For a fair visual comparison, we downsampled every synthetic dataset to 1219 cells (0.25×) in Supplementary Fig. 11a and observed that the read coverage of the pooled 1219 synthetic cells (pseudo-bulk) decreases as the synthetic cell number increases, an expected phenomenon reflecting the trade-off between the cell number and the per-cell sequencing depth given a fixed sequencing depth.

Supplementary Fig. 11b shows the synthetic datasets scReadSim generated with a fixed cell number but increasing sequencing depths 1.2$M$ (0.25×), 2.5$M$ (0.5×), 5.0$M$ (1×, the original sequencing depth), 10.1$M$ (2×), and 20.1$M$ (4×). As expected, the read coverage of the pooled cells (pseudo-bulk) increases as the sequencing depth increases.

Note that although scReadSim allows users to specify the sequencing depth arbitrarily, users must consider the biological limits so that the specified sequencing depth is reasonable (e.g., the total number of mRNA transcripts is the upper bound on the number of UMIs).

**Application 1: Benchmarking UMI deduplication tools**
For UMI-based scRNA-seq data, UMI deduplication tools have been developed to quantify gene expression levels from scRNA-seq reads, some of which may come from the same RNA molecule. UMI deduplication tools input scRNA-seq reads (containing UMIs and cell barcodes) and output a gene-by-cell UMI count matrix. scReadSim can be used for benchmarking UMI deduplication tools because it provides the ground-truth UMI count matrix in its synthetic scRNA-seq read generation process. Hence, we deployed scReadSim to the mouse 10x single-cell Multiome dataset[21] (the RNA-seq modality only) and used the synthetic reads to benchmark four state-of-the-art UMI

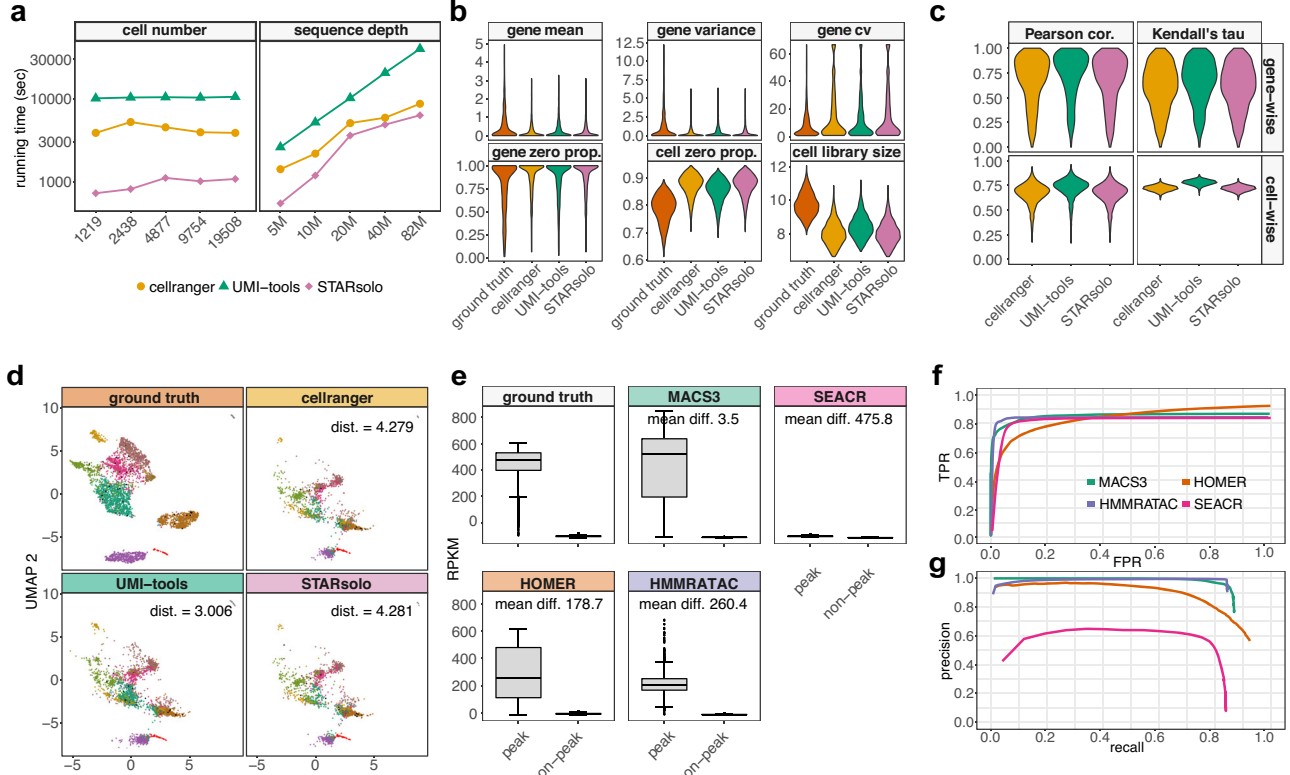

**Fig. 4 | Benchmark of UMI deduplication tools and peak calling tools using scReadSim's synthetic reads.** Benchmark of UMI deduplication tools (**a**–**d**). **a** Time usage of deduplication tools on synthetic datasets with varying cell numbers (at a fixed sequencing depth) or varying sequencing depths (at a fixed cell number). The y-axis indicates the time lapse (in seconds), and the x-axis shows the number of synthetic cells (left) or the total number of UMIs (sequencing depth, right). **b** Distributions of summary statistics of the UMI count matrices (ground truth, cellranger's output, UMI-tools' output, and STARsolo's output) at the gene level (mean, variance, coefficient of variance (cv), and zero proportion) and the cell level (zero proportion and library size). **c** Cell-wise and gene-wise correlations (Pearson correlation and Kendell's tau) between the ground-truth UMI count matrix and each deduplication tool's output UMI count matrix. **d** UMAP visualizations of the ground-truth UMI count matrix and each deduplication tool's output UMI count matrix. The mean value of the Euclidean distances of all synthetic cells ("Methods") is displayed for each UMI deduplication tool: a smaller value indicates

that the deduplicated UMI count matrix better agrees with the ground-truth UMI count matrix in UMAP visualization. Benchmark of peak calling tools (**e**–**g**). **e** Distributions of RPKM values of peak and non-peak regions in the ground truth (specified in scReadSim) and each tool's peak-calling result. The box center lines, bounds, and whiskers denote the medians, first and third quartiles, and minimum and maximum values within 1.5 × the interquartile range of the box limits, respectively. A difference of peak regions' mean RPKM (mean diff.) is calculated between the ground truth and each method's output. The numbers of peaks and non-peaks are as follows: ground-truth (2913 peaks and 2914 non-peaks), MACS3 (3310 peaks and 3290 non-peaks), SEACR (31,350 peaks and 31,339 non-peaks), HOMER (4782 peaks and 4780 non-peaks), and HMMRATAC (2726 peaks and 2571 non-peaks). **f, g** True positive rate (TPR) vs. false positive rate (FPR) curves (**f**) and precision vs. recall curves (**g**) using user-designed open chromatin regions as the ground-truth peaks.

deduplication tools: cellranger[11], STARsolo[12], UMI-tools[10], and Alevin[13]. Since Alevin is a transcript-level deduplication tool but scReadSim only allows gene-level ground-truth UMI counts in its current version (v.1.4.1), we will focus on cellranger, STARsolo, and UMI-tools in the following result comparison (Fig. 4a–d); the Alevin results are in Supplementary Figs. 12 and 13. Our benchmark study shows that UMI-tools achieves better accuracy than STARsolo and cellranger, while STARsolo is the most computationally efficient among the three tools.

First, in terms of computational efficiency, STARsolo runs faster than UMI-tools and cellranger on all synthetic datasets (with varying cell numbers and sequencing depths) (Fig. 4a). As expected, all three tools take longer to run when the sequencing depth increases, and their running time is unaffected by the cell number when the sequencing depth is fixed.

Second, the UMI count matrix output by UMI-tools agrees better with the ground-truth UMI count matrix in terms of (1) the distributions of six summary statistics, including four gene-level statistics (mean, variance, coefficient of variance, and zero proportion) and two cell-level statistics (zero proportion and library size) (Fig. 4b); and (2) Pearson correlations and Kendall's tau correlations between the

two matrices, both gene-wise (the correlation across cells per gene) and cell-wise (the correlation across genes per cell) (Fig. 4c).

Third, UMAP visualization shows that, UMI-tools outputs a UMI count matrix that is most similar to the ground-truth UMI count matrix (evidenced by the smallest average Euclidean distance between synthetic cells' two 2D UMAP coordinate vectors, calculated based on the ground-truth UMI counts and deduplicated UMI counts, respectively) and best preserves the separation among cell clusters (Fig. 4d).

**Application 2: Benchmarking scATAC-seq peak-calling tools**
For benchmarking scATAC-seq peak-calling tools, scReadSim provides ground-truth peaks and non-peaks by learning from user-specified trustworthy peaks and non-peaks or calling trustworthy peaks and non-peaks from real data.

Here we used scReadSim's synthetic scATAC-seq data (pseudo bulk), which contain designed ground-truth peaks and non-peaks ("Methods") and mimic the sci-ATAC-seq dataset[23], to benchmark four popular peak-calling tools: (1) MACS3, designed for identifying transcript factor binding sites from ChIP-seq data; (2) SEACR, designed for CUT&RUN data; (3) HOMER, designed for ChIP-seq data but also

widely used for ATAC-seq data; and (4) HMMRATAC, designed for ATAC-seq data.

Our benchmark results indicate that MACS3 and HMMRATAC have the best overall performance among the four peak-calling tools. In particular, the peaks called by MACS3 best resemble the ground-truth peaks in terms of read coverage, while HMMRATAC achieves the best balance between precision and recall.

First, we plotted the RPKM distributions in ground-truth peaks and non-peaks, as well as in the identified peaks and non-peaks by each peak-calling tool (Fig. 4e). The distributions indicate that MACS3 identifies the peaks that are the most similar to the ground-truth peaks, followed by HMMRATAC, HOMER, and SEACR. Specifically, the peaks called by HMMRATAC and SEACR are much longer than those by MACS3 and HOMER: the average peak lengths are 1543.0 bp and 1746.9 bp for HMMRATAC and SEACR, in contrast to 490.4 bp and 431.6 bp for MACS3 and HOMER. Possibly due to the design for CUT&RUN data, SEACR identifies the genomic regions with non-zero read coverages as peaks, so most of the peaks called by SEACR contain so few reads that these peaks' RPKMs are close to 0.

Second, all four tools detect most of the ground-truth peaks, under the assumption that a ground-truth peak is considered as correctly identified if a minimum length of the peak (i.e., half of the length of the shortest peak) is overlapped by an identified peak (Fig. 4f and Supplementary Figs. 14–15). Under the default settings of the four tools, the recall rates (i.e., the percentage of ground-truth peaks identified correctly) are all high: 86.8% for MACS3, 83.9% for SEACR, 92.4% for HOMER, and 84.4% for HMMRATAC.

Third, HMMRATAC achieves the highest precision rate of 90.3% (i.e., the percentage of identified peaks that are correct), followed by 76.5% for MACS3, 56.3% for HOMER, and 7.8% for SEACR. SEACR's low precision rate is possibly due to its design, leading to a vast number of identified peaks (34,834 peaks vs. 3678 peaks of MACS3, 5314 peaks of HOMER, and 3029 peaks of HMMRATAC) (Fig. 4g). Combining the precision and recall rates, the F1 scores indicate that HMMRATAC (87.3%) outperforms MACS3 (81.3%), HOMER (70.0%), and SEACR (14.3%).

## Discussion

scReadSim, a single-cell RNA-seq and ATAC-seq read simulator, generates realistic synthetic reads by mimicking real reads while providing ground truths including UMI counts for scRNA-seq and user-designed open chromatin regions for scATAC-seq. For both modalities, scReadSim allows the cell number and sequencing depth to vary. Although the current version of scReadSim uses scDesign2[6] as the internal count simulator (assuming that cells are from discrete cell types), scReadSim is easily adaptive to the new count simulator scDesign3[22], which allows cells to come from continuous trajectories or have spatial locations, accounts for conditions and batches, and has the capacity to generate single-cell multi-omics data. Moreover, scReadSim can be used for evaluating the robustness of read-level bioinformatics tools because it allows the generation of sequencing reads with varying levels of noise injected (e.g., sequencing errors, read positions, and read coverages).

Note that the current version of scReadSim (v.1.4.1) generates scRNA-seq reads with gene-level ground truths (UMI counts) only, not isoform-level ground truths including isoform abundance and alternative splicing events. Therefore, we do not recommend using the current version of scReadSim to benchmark computational tools that study isoforms or alternative splicing events. In addition, the default implementation of scReadSim may generate intronic synthetic reads when extracting sequences from the reference genome. Therefore, for benchmarking scenarios where intronic and exonic reads might make a difference, such as implementing cellranger (v7.0.0) under the default mode (counting intronic reads), we recommend using the transcriptome mode of scReadSim to generate exonic reads by

extracting sequences from a collapsed transcriptome, compiled from genes' annotated transcripts[10,25]. As a future direction, we will generalize scReadSim to generate scRNA-seq short and long reads with ground-truth RNA isoform abundance, a functionality unavailable in existing scRNA-seq simulators.

To generate multiple samples and conditions, the current version of scReadSim requires users to input one BAM file per sample or condition. As a future direction, we will leverage the count simulator scDesign3[22], which allows for adding sample and condition effects, to generate synthetic reads for multiple samples and conditions when the input is only one BAM file, by introducing sample and condition effects from realistic distributions learned from public data.

## Methods

Supplementary Note lists the software packages (with versions) used in scReadSim and the data analyses presented.

### The scReadSim method

The input of scReadSim is a BAM file storing the aligned sequencing reads from scRNA-seq or scATAC-seq. As an overview, scReadSim consists of three major steps: genome segmentation, synthetic count generation, and synthetic read generation.

The first step segments the genome so that each genomic region (segment) is defined as a feature. Specifically, for scRNA-seq, scRead-Sim requires users to input a gene annotation file to segregate the genome into genes and inter-genes; for scATAC-seq, scReadSim needs trustworthy peaks and non-peaks, which may be user-specified or not. If not user-specified, scReadSim provides two options; see the sub-section "scReadSim for scATAC-seq" for detail. Based on the trustworthy peaks and non-peaks, scReadSim defines the complementary genomic regions as gray areas. Next, scReadSim summarizes scRNA-seq reads in the input BAM file into a gene-by-cell UMI count matrix and an inter-gene-by-cell UMI count matrix, or scReadSim summarizes scATAC-seq reads in the input BAM file into a peak-by-cell count matrix and a non-peak-by-cell count matrix. We use a general term "feature-by-cell count matrix" to refer to each count matrix in the following text.

In the second step—synthetic count generation, scReadSim trains the count simulator scDesign2[6] (if the cells belong to distinct clusters; otherwise, scDesign3[22] can be used if the cells follow continuous trajectories) on the feature-by-cell count matrices to generate the corresponding synthetic count matrices. For scATAC-seq, scReadSim converts gray areas into non-peaks before generating the corresponding synthetic count matrix. In summary, scReadSim generates two synthetic count matrices—gene-by-cell and inter-gene-by-cell matrices—for scRNA-seq; scReadSim generates three synthetic count matrices—peak-by-cell, non-peak-by-cell, and non-peak-by-cell (converted from gray areas) matrices—for scATAC-seq.

The third step generates synthetic reads based on the synthetic count matrices, the input BAM file, and the reference genome (alternatively, for scRNA-seq, a collapsed transcriptome can be used to generate exon-only reads). The synthetic reads are outputted in a FASTQ or BAM file. Fig. 1a, b illustrate the scReadSim workflows for scRNA-seq and scATAC-seq, respectively.

**scReadSim for scRNA-seq.** The procedure below is for simulating data from UMI-based scRNA-seq technologies such as 10x Chromium[26] and Drop-seq[27]. For non-UMI-based scRNA-seq technologies such as Smart-seq2[28], the simulation procedure follows the subsection "scReadSim for scATAC-seq with user-designed open chromatin regions" with user-designed open chromatin regions to be replaced by genes.

1. Pre-processing real scRNA-seq data to obtain UMI count matrices
   To pre-process real scRNA-seq data for training, scReadSim requires a BAM file (containing scRNA-seq reads in cells) and a

gene annotation file (in GTF format). Based on the gene coordinates in the annotation file, scReadSim segregates the reference genome into two sets of features: genes and inter-genes. Specifically, to account for the overlapping genes in the annotation file, scReadSim first aggregates overlapping genes into non-overlapping regions (called "supergenes") to avoid double counting of reads from overlapping genes. Then scReadSim refers to non-overlapping genes and supergenes as "genes" and defines "inter-genes" as the complementary regions. (We acknowledge that this simple strategy, though avoiding double counting of reads, may obscure the read coverage differences of overlapping genes. As a future direction, by extending scReadSim to an isoform-level read simulator, we can solve this problem by providing the ground-truth isoforms and their abundance.) Given the defined genes and inter-genes, scReadSim counts the number of UMIs whose reads overlap with each gene in every cell to construct a gene-by-cell UMI count matrix, and similarly, an inter-gene-by-cell UMI count matrix. The two matrices contain the same cells in the same column order.

2.  Synthetic UMI count matrix generation

    To generate the gene- and inter-gene-by-cell UMI count matrices for synthetic cells, scReadSim trains the count simulator scDesign2[6] on the pre-processed gene- and inter-gene-by-cell UMI count matrices from real data, respectively. Note that scReadSim chooses scDesign2 because scDesign2 can generate realistic synthetic counts by capturing gene-gene correlations[9]; however, scReadSim is flexible to accommodate any scRNA-seq count simulator.

    Specifically, scDesign2 involves two steps: (1) model fitting on the real data count matrix and (2) generation of a synthetic count matrix by sampling synthetic cells from the fitted model. Using scDesign2, scReadSim generates synthetic gene-by-cell and inter-gene-by-cell count matrices from the real gene-by-cell and inter-gene-by-cell count matrices, respectively. Since scDesign2 requires cells to be in discrete cell types, scReadSim addresses this requirement by pre-clustering the cells using the gene-by-cell count matrix and Louvain clustering (in the Seurat R package[29]) if cells do not have predefined cell type labels. Note that scReadSim can also use the updated count simulator scDesign3[22] to generate the synthetic count matrices if cells are from continuous trajectories instead of discrete cell types.

    In summary, scReadSim generates the synthetic gene-by-cell and inter-gene-by-cell UMI count matrices in this step.

3.  Synthetic read generation

    To generate the scRNA-seq reads for synthetic cells, scReadSim uses:
    *   the synthetic gene-by-cell and inter-gene-by-cell UMI count matrices,
    *   the coordinates of the genes and inter-genes in the reference genome,
    *   the input BAM file,
    *   the reference genome,
    *   the user-specified read length (optional; default 90 nt).

    In UMI-based scRNA-seq data, every paired-end read has only one end (referred to as the read 2) containing the RNA sequence information, while the other end (referred to as the read 1) contains the cell barcode and UMI information[18,30]. The generation of these two ends is described below. (The two ends are shown as concatenated in Fig. 1a for the ease of visualization, but they are not together in one sequence.)

    (a)  Generation of the read 2 (RNA sequence)

    For simplicity, we use a "read" to refer to the read 2. For each feature (gene or inter-gene) $i$ in one synthetic cell (as an example), scReadSim (1) extracts the corresponding UMI count $u_i$ (i.e.,

the number of synthetic UMIs) from the synthetic feature-by-cell UMI count matrices; (2) summarizes the read count of each UMI from the real reads that overlap with this feature $i$, obtaining the empirical distribution $\widehat{F}_i$ of the read count per UMI; (3) samples the number of synthetic reads $c_{ij}$ for each synthetic UMI $j = 1, …, u_i$ according to the empirical distribution $\widehat{F}_i$; and (4) for the $j$-th synthetic UMI,

i.  samples (with replacement) a real UMI from the real UMIs with read counts equal to $c_{ij}$;

ii.  samples (with replacement) $c_{ij}$ real reads from the reads belonging to the sampled real UMI;

iii.  generates $c_{ij}$ synthetic reads from the $c_{ij}$ real reads, given the read length and the reference genome;

iv.  assigns the $c_{ij}$ synthetic reads to the $j$-th synthetic UMI.

Specifically, in (4), scReadSim converts the $c_{ij}$ real reads' 5′ positions in the reference genome into the $c_{ij}$ synthetic reads' 5′ positions (after adding random shifts, detailed in the next paragraph); then scReadSim finds the $c_{ij}$ synthetic reads' 3′ positions in the reference genome based on the read length (e.g., if a synthetic read has the 5′ position $x$ and the read length $l$, then its 3′ position is $x + l - 1$). Next, given every synthetic read's 5′ and 3′ positions, and the strand information (preserved from the real reads), scReadSim extracts the synthetic read sequence from the reference genome sequence, obtaining a total of $c_{ij}$ read sequences.

Regarding the random shifts mentioned in the above paragraph, scReadSim adds a random shift (sampled uniformly from −4 to 4 nt) to each real read's 5′ position to obtain a synthetic read's 5′ position. The purpose of doing so is to resolve duplicate 5′ positions due to the sampling with replacement. To make the synthetic reads more realistic, scReadSim also allows the introduction of substitution errors, as described in the subsection "Substitution errors introduced into synthetic reads."

Alternatively, in (4), scReadSim enables users to generate scRNA-seq synthetic reads from the transcriptome. To do so, scReadSim requires a collapsed transcriptome FASTA file. This collapsed transcriptome can be generated from a reference genome and a gene annotation GTF/GFF file by (i) combining all transcripts of a gene into a collapsed transcript with the software cgat[31]; (ii) generating the transcriptome FASTA file based on the reference genome and the collapsed transcript annotations using the software gffread[32]. For each gene $i$'s synthetic UMI $j = 1, …, u_i$, scReadSim uniformly samples the $c_{ij}$ synthetic reads' 5′ positions from $[0, l_i]$, where $l_i$ is gene $i$'s collapsed transcript length; then scReadSim finds the $c_{ij}$ synthetic reads' 3′ positions in the collapsed transcript based on the read length and the boundary at $l_i$ (that is, the 3′ positions cannot exceed the boundary, so some reads may be shorter than the specified read length). Next, given every synthetic read's 5′ and 3′ positions in the collapsed transcript, scReadSim extracts the read sequence from the collapsed transcript sequence, resulting in a total of $c_{ij}$ read sequences.

(b)  Generation of the read 1 (cell barcode and UMI)

The read 1 is a concatenated sequence string of a UMI and a cell barcode. Accordingly, scReadSim concatenates a randomly generated cell barcode and a UMI to form a synthetic read 1, which will then be paired with a synthetic read 2 generated above. Below we describe the generation of a cell barcode and a UMI for each synthetic read 1, whose length is specified as 26 nt (including a 16-nt cell barcode and a 10-nt UMI).

Specifically, for each synthetic cell, scReadSim generates a 16-nt cell barcode by randomly sampling A, C, G, and T with replacement for 16 times and assigns this cell barcode to all synthetic reads belonging to the synthetic cell. To generate a UMI

for each feature (gene or inter-gene) $i$ in each synthetic cell, scReadSim (1) extracts the corresponding UMI count $u_i$ (i.e., the number of UMIs) from the synthetic feature-by-cell UMI count matrices; (2) generates $u_i$ 10-nt UMIs, with each UMI created by randomly sampling A, C, G, and T with replacement for 10 times; and (3) assigns the $u_i$ UMIs to the synthetic reads 2 based on the read-UMI relationship generated in (a): assigning each synthetic UMI $j$, $j = 1, …, u_i$, to $c_{ij}$ reads 2. Note that the lengths of cell barcodes and UMIs can be user-specified (default lengths are 16 and 10 nt, respectively).

In summary, scReadSim outputs paired-end synthetic reads in two FASTQ files for reads 1 and reads 2 separately. Regarding the Phred quality scores in the FASTQ files, the read 2 FASTQ file uses the quality score 37 for reference sequences and the quality score 24 for erroneous substitutions, while the read 1 FASTQ file uses the quality score 37 for all sequences. Optionally, scReadSim can output a BAM file, which contains the mapped reads from the two FASTQ files to the reference genome.

## scReadSim for scATAC-seq.

1. Pre-processing

   To pre-process real scATAC-seq data for training, scReadSim requires a BAM file (containing scATAC-seq reads in cells) and user-specified trustworthy peaks and non-peaks about the input BAM file. Alternatively, if users do not specify trustworthy peaks and non-peaks, scReadSim provides users with two options for generating trustworthy peaks and non-peaks. Option One is that scReadSim first deploys the peak-calling tool MACS3[14] to identify the trustworthy peaks using a stringent rule (by setting the q-value threshold to 0.01) and the trustworthy non-peaks using another stringent rule (the non-peaks are complementary to the peaks called under the q-value threshold of 0.1). Option Two allows users to input a set of possible peaks from public databases, for instance, the ENCODE cCREs (candidate cis-regulatory elements)[33] summarized from various DNA accessibility assays and ChIP-seq experiments. This input set of possible peaks does not depend on the input BAM file, similar to the input gene annotation scReadSim uses to generate scRNA-seq reads. To define the trustworthy peaks and non-peaks under this Option Two, scReadSim selects the input possible peaks and non-peaks (defined as the regions complementary to the input possible peaks) that overlap with the peaks and non-peaks identified by MACS3 in Option One. The definition of overlapping is, if an input possible peak (or non-peak) has half of its length covered by one or more MACS3-identified peaks (or non-peaks), then such an input possible peak (or non-peak) would be considered trustworthy.

   Based on the trustworthy peaks and non-peaks, scReadSim defines the "gray areas" as the genomic regions complementary to the peaks and non-peaks, and such gray areas represent the chromatin regions that cannot be confidently classified as peaks or non-peaks. In summary, scReadSim segments the reference genome into three sets of features: peaks, non-peaks, and gray areas (Fig. 1b). Then scReadSim counts the number of reads overlapping each peak in every cell to construct a peak-by-cell count matrix. scReadSim also generates a non-peak-by-cell count matrix similarly. The two matrices contain the same cells in the same column order.

2. Synthetic count matrix generation

   In scATAC-seq read generation, scReadSim provides ground-truth peaks and non-peaks for benchmarking purposes by learning from user-specified trustworthy peaks and non-peaks or calling trustworthy peaks and non-peaks from real data. To generate the peak-by-cell and non-peak-by-cell count matrices

for synthetic cells, scReadSim trains the count simulator scDesign2 on the pre-processed peak-by-cell and non-peak-by-cell count matrices from real data. Next, scReadSim converts the gray areas into non-peaks (so that the peaks can be regarded as "ground-truth peaks") and constructs a synthetic count matrix based on the gray areas' lengths, trustworthy non-peaks' lengths, and the already-generated synthetic non-peak-by-cell count matrix.

Specifically, scReadSim generates gray areas' synthetic read counts based on the trustworthy non-peaks' synthetic read counts so that the gray areas' synthetic read coverage mimics the non-peaks' synthetic read coverage. For each gray area,

(a) scReadSim first finds the closest non-peak on the 5' of the gray area and extracts the non-peak's synthetic count vector (the non-peak's corresponding row) from the synthetic non-peak-by-cell count matrix;

(b) scReadSim then randomly masks (sets to 0) the count entries of the non-peak's synthetic count vector with a probability equal to $\max\left(1 - \frac{\text{length of gray area}}{\text{length of non-peak}}, 0\right)$ and assigns the masked count vector as the gray area's synthetic count vector. This random masking essentially scales the non-peak's synthetic read coverage according to the ratio of the gray area's length and the non-peak's length (when the gray area is shorter than the non-peak) so that the gray area's synthetic read coverage mimics that of its nearby non-peak. When the gray area is longer than the non-peak, the gray area's read counts are set to the non-peak's read counts, so the gray area has a lower read coverage and becomes more like a non-peak.

3. Synthetic read generation

   To generate the scATAC-seq reads for synthetic cells, scReadSim uses
   - the synthetic peak-by-cell and non-peak-by-cell count matrices,
   - the coordinates of the peaks and non-peaks in the reference genome,
   - the input BAM file,
   - the reference genome,
   - the user-specified read length (optional; default 50 nt).

As scATAC-seq usually generates paired-end reads, scReadSim accordingly generates synthetic paired-end reads. For clarity, we use a "read" to denote one end in a paired-end read. For each peak or non-peak in each synthetic cell, scReadSim (1) extracts the corresponding count $c$ (i.e., the number of synthetic reads) from the synthetic feature-by-cell (a feature refers to a peak or non-peak) count matrices; (2) samples (with replacement) $\lceil c/2 \rceil$ (rounding $c/2$ to the least integer greater than or equal to $c/2$) real reads that overlap this feature in the BAM file; (3) finds the mates of (i.e., real reads in pairs with) the $\lceil c/2 \rceil$ sampled real reads from the BAM file; and (4) generates $\lceil c/2 \rceil$ synthetic read pairs based on the $\lceil c/2 \rceil$ real read pairs, the read length, and the reference genome.

Specifically, in (4), scReadSim follows a procedure similar to the generation of synthetic read 2 for scRNA-seq. The only difference is the accommodation of paired-end reads. In the random shifting step to avoid duplicate reads, scReadSim adds the same random shift to a pair of reads' 5' positions to keep the fragment size unchanged after the shift.

Typically, a scATAC-seq BAM file stores cell barcodes as additional information. To generate a cell barcode (e.g., 16-nt long), scReadSim randomly samples A, C, G, and T with replacement 16 times. Then scReadSim assigns a unique cell barcode to all synthetic reads of each synthetic cell.

**scReadSim for scATAC-seq with user-designed open chromatin regions.**

1. Pre-processing

   Optionally, scReadSim allows users to design ground-truth open chromatin regions and then generates synthetic scATAC-seq reads accordingly. When users take this option, scReadSim requires users to input a BAM file, the trustworthy peaks and non-peaks identified from the BAM file (user-specified or scReadSim-defined if unspecified), and the user-designed ground-truth peaks (Supplementary Fig. 16). Given the user-designed ground-truth peaks, scReadSim specifies the regions complementary to the ground-truth peaks as the ground-truth non-peaks. In summary, scReadSim defines two sets of peaks and non-peaks: (1) the set of "trustworthy peaks and non-peaks" based on the user-specified (or scReadSim-defined) trustworthy peaks and non-peaks and (2) the set of "ground-truth peaks and non-peaks" based on the user-designed ground-truth peaks.

   For the "trustworthy peaks and non-peaks" feature set, scReadSim counts the number of reads overlapping each trustworthy peak or non-peak in every cell to construct a trustworthy-peak-by-cell count matrix and a trustworthy-non-peak-by-cell count matrix. The two matrices contain the same cells in the same column order.

   For the "ground-truth peaks and non-peaks" feature set, since the specified ground-truth peaks may not correspond to any real peaks in the BAM file, scReadSim constructs the ground-truth peaks' counts by mapping the trustworthy peaks to the "most similar" ground-truth peaks in terms of region length. Below are the detailed construction steps.

   (a) For every ground-truth peak, scReadSim defines a corresponding trustworthy peak as follows. First, scReadSim finds the ground-truth peak's 50 most similar trustworthy peaks (in the same chromosome) in terms of lengths (Supplementary Fig. 17a). Second, scReadSim chooses the trustworthy peak with the largest ratio of read count to peak length.

   (b) To construct the ground-truth-peak-by-cell count matrix, for every ground-truth peak, scReadSim adds the corresponding trustworthy peak's row in the trustworthy-peak-by-cell count matrix to the ground-truth-peak-by-cell count matrix (Supplementary Fig. 17b). This constructed ground-truth-peak-by-cell count matrix has the same column order as the trustworthy-peak-by-cell count matrix. The ground-truth-non-peak-by-cell matrix is constructed similarly, where every ground-truth non-peak is mapped to a trustworthy non-peak. Note that for each ground-truth non-peak, scReadSim first finds the 50 most similar trustworthy non-peaks in terms of lengths, and then it chooses the trustworthy non-peak with the smallest ratio of read count to peak length.

   The ground-truth-peak-by-cell and ground-truth-non-peak-by-cell count matrices, as well as the maps from the ground-truth features to the trustworthy features, will be used in the next steps.

2. Synthetic count matrix generation

   Similar to the synthetic count matrix generation for scATAC-seq without user-designed ground-truth peaks (step "2. Synthetic count matrix generation" in the section "scReadSim for scATAC-seq"), scReadSim trains scDesign2 on the ground-truth-peak-by-cell and ground-truth-non-peak-by-cell count matrices (constructed from step "1. Pre-processing") to generate the ground-truth-peak-by-cell and ground-truth-non-peak-by-cell count matrices for synthetic cells.

3. Synthetic read generation

   To generate synthetic scATAC-seq reads from the synthetic ground-truth-peak-by-cell and ground-truth-non-peak-by-cell

count matrices, for each ground-truth feature (peak or non-peak) in every synthetic cell, scReadSim (1) extracts the corresponding count $c$ (i.e., the number of synthetic reads) from the synthetic ground-truth-feature-by-cell count matrices; (2) samples (with replacement) $\lceil c/2 \rceil$ real reads in the BAM file that overlap the ground-truth feature's corresponding trustworthy feature; (3) obtain the mates of the $\lceil c/2 \rceil$ sampled real reads; and (4) generates $\lceil c/2 \rceil$ synthetic read pairs based on the $\lceil c/2 \rceil$ real read pairs, the read length, the signed distance between the ground-truth feature and its corresponding trustworthy feature (based on the two features' genomic coordinates), and the reference genome.

Specifically, in (4), scReadSim converts the $c$ real reads' 5′ positions in the reference genome to the $c$ synthetic reads' 5′ positions after two adjustments: first, shifting real reads from the trustworthy feature to the ground-truth feature based on the signed distance between the two features; second, adding a random shift to each real read pair's 5′ positions. In greater detail, in the first adjustment step, assuming that the signed distance between the trustworthy feature and the ground-truth feature is $+d$, if a real read has a 5′ position $x$, then the synthetic read's 5′ position is $x + d$. This adjustment ensures that the synthetic reads are located in the ground-truth feature as how the real reads are located in the trustworthy feature. The second adjustment (random shift addition) is to avoid duplicate reads due to sampling with replacement. Specifically, scReadSim adds the same random shift to a pair of reads' 5′ positions to keep the fragment size unchanged after the shift.

**scReadSim for single-cell multi-omics.** The multi-modality of scReadSim enables it to simulate single-cell multi-omics (simultaneously measured scRNA-seq and scATAC-seq modalities) sequencing reads. In order to do so, scReadSim requires users to input a single-cell multi-omics dataset and the corresponding reference genome. Following the procedures for simulating individual scRNA-seq and scATAC-seq modalities described above, scReadSim first processes the two modalities to construct real count matrices, including scRNA-seq gene-by-cell and inter-gene-by-cell UMI count matrices and scATAC-seq peak-by-cell and non-peak-by-cell read count matrices. Then, given these count matrices, scReadSim implements scDesign3[22], a single-cell multi-omics count simulator, to simulate synthetic cells' UMI and read counts for both modalities. Finally, based on the synthetic counts, scReadSim generates synthetic reads for both modalities in the FASTQ format, along with a list of synthetic cell barcodes. As a result, the synthetic single-cell multi-omics sequencing reads would contain synthetic cells with both modalities and ground truths, including UMI abundance for the scRNA-seq modality and open chromatin regions for the scATAC-seq modality.

**scReadSim for multiple samples/conditions.** scReadSim provides a module for users to simulate single-cell sequencing reads from multiple samples or conditions. It requires users to input multiple BAM files corresponding to multiple samples or conditions, as well as the corresponding reference genome file. Then, scReadSim trains a model for each input BAM file and simulates synthetic reads separately for each sample or condition. Specifically for scATAC-seq, to provide comparable ground-truth peaks across the multiple synthetic samples, scReadSim requires users to input a set of possible peaks, such as the ENCODE cCREs (candidate cis-regulatory elements)[33]. This pre-specified peak set provides scReadSim with a consistent segmentation of the reference genome, and scReadSim's ground-truth peaks will be defined based on this segmentation.

**Doublet/multiplet detection.** In single-cell sequencing technologies that use droplets or wells (called reaction volumes) to isolate single

cells, doublets/multiplets exist when more than one cell is captured by the reaction volume and mistaken as a single cell. Doublets/multiplets would bias downstream data analysis[34–36]. As scReadSim mimics real data, it requires doublets/multiplets to be removed from real data so its synthetic data would not contain doublets/multiplets. To remove doublets/multiplets from real data, scReadSim uses scDblFinder[37] as the doublet detection method based on the benchmark study[36]. For scRNA-seq data, scDblFinder is applied to the real gene-by-cell UMI count matrix to identify the columns ("cells") that likely correspond to doublets/multiplets; then, the identified columns would be removed from both the real gene-by-cell and inter-gene-by-cell UMI count matrices. For scATAC-seq data, scDblFinder is applied to the real peak-by-cell read count matrix to identify the columns ("cells") that likely correspond to doublets/multiplets; then, the identified columns would be removed from the real peak-by-cell and non-peak-by-cell read count matrices. Finally, after removing the doublets/multiplets from real data, scReadSim is trained on the post-removal real count matrices to generate synthetic data.

**Output in a FASTQ or BAM format.** To output the synthetic reads (excluding the UMI-based scRNA-seq reads 1, which contain cell barcodes and UMIs) in a FASTQ or BAM file, scReadSim first records the coordinates of the synthetic reads in a BED file. Then scReadSim uses the bedtools[38] `getfasta` function to extract every synthetic read's sequence from the reference genome and store the synthetic reads' sequences in a FASTA file, which is then converted to a FASTQ file by the seqtk software[39]. For paired-end reads, scReadSim generates two FASTQ files: one for the read 1 and the other for the read 2. Users may choose FASTQ as the output format of scReadSim, or they may additionally output a BAM file by aligning the synthetic reads to the reference genome by the alignment software bowtie2[40]. In addition, the list of synthetic cell barcodes is outputted for users' reference, which could serve as a cell barcode white list for downstream analyses.

**Substitution errors introduced into synthetic reads.** To mimic the sequencing error rates observed in Illumina sequencing reads, scReadSim introduces substitution errors into the synthetic reads. Specifically, scReadSim first uses the software fgbio[41] to calculate every position $i$'s average error rate (i.e., the probability that the position is sequenced as a wrong nucleotide, denoted by $p_i$) and three substitution error rates (i.e., the probabilities that the position is wrongly sequenced as the three other nucleotides; the sum of the three probabilities is $p_i$) based on all real reads. Then for each synthetic read, scReadSim decides if every base call, say at position $i$, is erroneous by sampling from Bernoulli $(p_i)$. If the base call is decided to be erroneous (i.e., the Bernoulli sample is 1), a substitution nucleotide is randomly sampled from the other three nucleotides with probabilities proportional to the three substitution error rates at position $i$. Specifically, for paired-end sequencing technologies like scATAC-seq, scReadSim introduces substitution errors separately for the two mates in a pair.

### Data analysis
**Count-level comparison between synthetic data and real data.** To verify that scReadSim generates synthetic data that mimics real data at the read-count level, we compared synthetic and real gene-by-cell UMI count matrices in the following aspects: summary statistics, correlations among top-expressed genes (or peaks), and two-dimensional (2D) visualizations. For scATAC-seq, in addition to these three aspects used in scRNA-seq, we compared the RPKM distributions between synthetic and real peak-by-cell read count matrices.

- Summary statistics. We utilized the following summary statistics: for every feature, we calculated the mean, variance, coefficient of variance, and zero proportion across cells; for every cell,

we calculated the library size and zero proportion across features.

- Feature-feature correlations. To illustrate that scReadSim preserves the correlations among the features (genes for scRNA-seq; peaks for scATAC-seq), we selected the 100 top-expressed features from the real count matrix and used a heatmap to show the correlations among these features in the real and synthetic count matrices, respectively.

- 2D visualizations. We utilized principal component analysis (PCA) and uniform manifold approximation and projection (UMAP)[42,43] to obtain the 2D embeddings of cells in the combined real and synthetic count matrix: we first used PCA to obtain the 30 principal components (PCs) and then deployed UMAP to obtain the 2D embeddings. We labeled the cells with the dataset sources (synthetic cells generated by scReadSim or real cells). We also reported the median of the integration local inverse Simpson's index (miLISI) proposed by Korsunsky et al.[44] to quantify the mixing level of the synthetic cells and the real cells (see the subsection "Evaluation metrics"). The miLISI value (ranging from 1 to 2) is close to 2 if the synthetic cells and real cells are perfectly mixed.

- RPKM. We used the quantity RPKM (see the subsection "Evaluation metrics") to summarize the read coverage in a genomic region. Specifically, we reported the RPKM values separately for the peaks and non-peaks.

**Sequence-level comparison between synthetic scRNA-seq data and real scRNA-seq data.** We deployed scReadSim onto the mouse 10x single-cell Multiome dataset[21] (the RNA-seq modality only) by inputting the reference genome, the BAM file, and the gene annotation GTF file (Fig. 1a). Then, scReadSim generated synthetic scRNA-seq reads in the FASTQ and BAM formats.

To show that scReadSim's synthetic scRNA-seq data mimics real data at the read-sequence level, we compared the synthetic BAM file and real BAM file in terms of the k-mer spectrum, the error rate per base, and the genome browser visualization.

- The k-mer spectrum. The k-mer spectrum measures the occurrences of k-mers (i.e., substrings of length $k$) within the read sequences. We used the tool Jellyfish[45] with the default settings to obtain the 11-mer, 21-mer, 31-mer, and 41-mer spectra in the synthetic reads and real reads, respectively.

- Error rate per base. We calculated and compared the substitution error rate for every base in the synthetic reads and real reads, respectively. The substitution error rate per base within reads was obtained using the software fgbio[41] with the function `ErrorRateByReadPosition` by setting the option `collapse` as false.

- Genome browser visualization. We used the IGV genome browser[46] to visualize the real and synthetic BAM files. This visualization illustrates the pseudo-bulk-level similarity of read coverages between the two BAM files.

**Sequence-level comparison between synthetic scATAC-seq data and real scATAC-seq data.** We deployed scReadSim onto the sci-ATAC-seq and mouse 10x single-cell Multiome dataset[21] (the ATAC-seq modality only) by inputting the reference genome, the BAM file, and the trustworthy peaks and non-peaks (Fig. 1b). The trustworthy peaks were identified by MACS3 from the input BAM file with a stringent rule (by setting the q-value parameter as `-q 0.01`). To obtain the trustworthy non-peaks, we first deployed MACS3 onto the input BAM file with a relaxed rule (by setting the q-value parameter as `-q 0.1`) to identify peaks. Then we took the inter-peaks as the trustworthy non-peaks. Lastly, we used scReadSim to generate synthetic scATAC-seq reads in the FASTQ and BAM formats.

To verify that scReadSim simulates realistic scATAC-seq data at the read-sequence level, we compared the synthetic BAM file and

the real BAM file in terms of the k-mer spectrum, the error rate per base, the genome browser visualization (see the subsection "Sequence-level comparison between synthetic scRNA-seq data and real scRNA-seq data"), and, additionally, the fragment size distribution. We further verified the realistic nature of the synthetic reads by deploying the peak caller MACS3 onto both the synthetic BAM file and the real BAM file and then evaluating the consistency between the two sets of called peaks.

- Fragment size distribution. For paired-end sequencing reads, the fragment size distribution is the distribution of the distance between the two ends of a read pair. We used the function `bamPEFragmentSize` of the software deeptools[47] with the default settings to obtain the fragment size distributions of synthetic scATAC-seq reads and real scATAC-seq reads, respectively.
- Called peaks. To verify that the synthetic reads mimic the real reads, we deployed the peak-calling tool MACS3 with the same setting (by setting the q-value parameter as `-q 0.01`) onto the real and synthetic BAM files and compared the two sets of called peaks using a Venn diagram. Moreover, we calculated the receiver operating characteristic (ROC) curve and the precision-recall curve by treating the real-data peaks called at `-q 0.01` as the ground truths. Then we first applied MACS3 to the synthetic reads with a relaxed criterion to obtain a long list of synthetic-data peaks (by setting the q-value parameter as `-q 0.5`) and next used various q-value thresholds for further pruning of the synthetic-data peaks. To match the synthetic peaks to the real-data peaks, we defined a match if a synthetic-data peak overlaps with a real-data peak for at least half of the minimal length of real-data peaks. Hence, given a q-value threshold and the corresponding set of synthetic-data peaks, we calculated the true positive rate and false positive rate for the ROC curve, and the precision and recall for the precision-recall curve.

**Benchmark of UMI deduplication tools.** We deployed scReadSim with the default setting (generating scRNA-seq reads from the genome instead of the collapsed transcriptome; see the subsection "scReadSim for scRNA-seq") to the mouse 10x single-cell Multiome dataset[21] (the RNA-seq modality only) to generate the synthetic scRNA-seq data. Then we used scReadSim's synthetic scRNA-seq reads as the input data and scReadSim's gene-by-cell UMI count matrix as the ground truth to benchmark four UMI deduplication tools: UMI-tools, STARsolo, and cellranger (gene-level), as well as Alevin (transcript-level). We applied each deduplication tool to scReadSim's synthetic scRNA-seq reads to obtain a deduplicated UMI count matrix. All four deduplication methods only count exonic reads (cellranger was implemented by setting the argument `include-introns` as false). Since these deduplication tools may output different numbers of cells, we only considered the common cells outputted by all the tools for a fair comparison. We also removed the genes with zero expression levels in all cells in any deduplicated UMI count matrices or the ground-truth UMI count matrix. We compared the deduplicated UMI count matrices with the ground-truth count matrix in the following three aspects.

1. We compared the UMI count matrices using the distributions of six summary statistics, including four gene-level statistics (mean, variance, coefficient of variance, and zero proportion) and two cell-level statistics (zero proportion and cell library size). The results are in Fig. 4b.
2. We calculated associations between the ground-truth UMI count matrix and each deduplicated UMI count matrix, gene-wise and cell-wise (Fig. 4c). The association measures include the Pearson correlation and Kendall's tau.
3. We used PCA and UMAP for dimension reduction and visualization to examine whether the deduplicated UMI count matrices preserve the 2D cell embeddings of the ground-truth UMI count

matrix. Cells are colored by the cell clusters outputted by scReadSim (the clusters are from the real data used to train scReadSim). Specifically, we first log-transformed the three UMI count matrices. Then we performed PCA on the ground-truth matrix to obtain the top 30 PCs and the associated 30 loading vectors. Next, we applied UMAP to the top 30 PCs to find the 2D cell embeddings. For the deduplicated matrices, we first projected their cells to the same 30-dimensional PC space by left multiplying the 30-by-$p$ loading matrix (with rows as the top 30 loading vectors; $p$ is the number of genes); second, we projected the deduplicated matrices' cells from the PC space to the same UMAP space using the `predict()` function in the R package umap. We computed the Euclidean distance between each synthetic cell's two UMAP coordinates, calculated based on the cell's ground-truth UMI counts and deduplicated UMI counts, respectively. The mean value of the Euclidean distances of all synthetic cells is displayed for each UMI deduplication tool: a smaller value indicates that the deduplicated UMI count matrix better agrees with the ground truth UMI count matrix in the UMAP visualization. The results are in Fig. 4d.

In addition, we measured the time complexity by deploying each deduplication tool to scReadSim's synthetic data with varying cell numbers (Fig. 4a): 1219 (0.25×), 2438 (0.5×), 4877 (1×, the original cell number), 9754 (2×), 19508 (4×); and varying sequencing depth 1.2$M$ (0.25×), 2.5$M$ (0.5×), 5.0$M$ (1×, the original sequencing depth), 10.1$M$ (2×), 20.1$M$ (4×). The analysis was implemented on a server with the 256x Intel®Xeon Phi™ CPU 7210 at 1.30 GHz.

**Benchmark of peak-calling tools.** To generate synthetic scATAC-seq reads with specified ground-truth peaks (i.e., open chromatin regions), we provided scReadSim with three inputs: the mouse sci-ATAC-seq BAM file[23], the trustworthy peaks and non-peaks called from the BAM file by MACS3, and a list of specified ground-truth peaks (Supplementary Fig. 16). In detail, we specified each ground-truth peak as [TSS, TSS + LEN − 1], where TSS is a gene's transcription start site, and the peak length LEN is uniformly sampled from 250 bp to 550 bp. If any ground-truth peaks were overlapping, we merged them into one peak so that the final ground-truth peaks were non-overlapping. Taking the three inputs, scReadSim generated synthetic scATAC-seq reads in FASTQ and BAM files (see the subsection "scReadSim for scATAC-seq with user-designed open chromatin regions").

To benchmark four peak-calling tools—MACS3, HOMER, SEACR, and HMMRATAC, we deployed these tools with their default settings onto the synthetic scATAC-seq BAM file. Specifically, MACS3 was run under the `-BAMPE` mode with the q-value threshold 0.05; HOMER was run under the `-region` mode with the parameter `-minDist 150`; SEACR was run under the `-stringent` mode with the control threshold 0.5; HMMRATAC was run under the default setting. Then we compared the peaks called by each tool with the ground-truth peaks used to generate the synthetic data. Our comparison consisted of the following three aspects:

- First, we compared the distributions of RPKM values of the called peaks and the ground-truth peaks, and we calculated the mean difference between the two distributions. We also compared the distributions of RPKM values of the called non-peaks and the ground-truth non-peaks. The results are in Fig. 4e.
- Second, treating the ground-truth peaks as the truths, we plotted the ROC curve (Fig. 4f) and the precision-recall curve (Fig. 4g) for each tool by applying varying thresholds to the called peaks. We considered a called peak as correct if it overlapped at least 225 bp of any ground-truth peak, where 225 bp is half the length of the shortest ground-truth peak.
- Third, we directly compared the four sets of peaks called by the four tools using the Venn diagram and the upset plot, plotted by

the Python package Intervene[48]. The results are in Supplementary Figs. 14–15.

**Comparison of scReadSim with minnow.** We first deployed scReadSim to the mouse 10x single-cell Multiome dataset[21] (the RNA modality only) to generate synthetic reads in a BAM file. Since minnow does not accept reads but only a gene-by-cell UMI count matrix as input, we inputted into minnow the synthetic gene-by-cell UMI count matrix (generated by scReadSim as an intermediate step) and kept minnow's parameters as default. The synthetic reads minnow generated were outputted in two FASTQ files (for the two ends of paired-end reads), and we aligned them to the same reference genome scReadSim used. After binning the reference genome (chromosome 1) into 1000-bp windows, we plotted the read coverage per window for the real data, scReadSim's synthetic data, and minnow's synthetic data (Fig. 2a). We further calculated two correlations (Spearman correlation and Pearson correlation) of read coverage between the real and synthetic data across the windows. We also provided a closer view of the read coverage (in the region chr1:86,425,858–86,443,378) using the IGV genome browser (Fig. 2a inset).

**Comparison of scReadSim with SCAN-ATAC-Sim.** We used the sci-ATAC-seq dataset[23] to train scReadSim and SCAN-ATAC-Sim. Then we compared the two simulators' performance in preserving the peak counts in real data.

For scReadSim, we inputted the real dataset's BAM file, the reference genome, and the trustworthy peaks (called at the q-value threshold 0.01) and non-peaks (complementary to the peaks called at the q-value threshold 0.1) identified by MACS3 from the BAM file. For each cell type, we used scReadSim to generate the same number of synthetic cells as the number of real cells. Moreover, we recorded the trustworthy peaks.

Since SCAN-ATAC-Sim requires per-cell-type input, we focused on the four largest cell types containing the most cells (predefined in ref. 23, including Hematopoietic progenitors, Erythroblasts, Monocytes, and Immature B cells). Hence, we only examined the real reads and scReadSim's synthetic reads for the four cell types.

For SCAN-ATAC-Sim to generate scATAC-seq data for the four cell types, it requires the four cell types' pseudo-bulk ATAC-seq BAM files and their corresponding peak lists as input. Hence, we first extracted the four cell types' reads from the real data BAM file to construct the four corresponding per-cell-type BAM files. Then we used MACS3 to call peaks from each per-cell-type BAM file. Finally, we inputted the four per-cell-type peak lists along with the four per-cell-type BAM files into SCAN-ATAC-Sim. For SCAN-ATAC-Sim's parameters, we specified the cell number of each cell type to the real cell number, which is also scReadSim's synthetic cell number, of that cell type. For the other parameters, we used the default values suggested in the SCAN-ATAC-Sim paper[20]. SCAN-ATAC-Sim does not output synthetic read sequences but only a BED file containing synthetic reads' starting and ending positions, along with the reads' cell IDs.

To evaluate scReadSim and SCAN-ATAC-Sim's performance at the peak-count level, we constructed scReadSim and SCAN-ATAC-Sim's peak-by-cell read count matrices (one for each simulator) from scReadSim's output BAM file and SCAN-ATAC-Sim's output BED file, by counting each simulator's synthetic reads that overlap with the trustworthy peaks in the synthetic cells. We also constructed a real count matrix in the same way, so the three count matrices have the same dimensions. In three aspects, we compared the three peak-by-cell count matrices (real, scReadSim, and SCAN-ATAC-Sim).

First, we calculated the distributions of six summary statistics based on the entries of each count matrix: peak-wise mean, variance, coefficient of variance, and zero proportion across cells, as well as cell-wise library size and zero proportion across peaks. We further performed the two-sample Kolmogorov-Smirnov (KS) test to compare each summary statistic's distributions in the real count matrix and a synthetic count matrix (scReadSim or SCAN-ATAC-Sim) to measure the resemblance of the synthetic count matrix to the real count matrix (a smaller KS statistic value indicates greater similarity). The results are in Supplementary Fig. 6a.

Second, we obtained the 2D UMAP cell embeddings of the real and synthetic count matrices using the following steps. The results are in Fig. 2b.

1. We transformed the count matrices by applying the Term Frequency–Inverse Document Frequency (TF-IDF) transformation followed by the logarithmic transformation.
2. We performed PCA on the real pre-processed matrix to obtain the top 30 PCs and the associated 30 loading vectors. Next, we applied UMAP to the top 30 PCs to find the 2D cell embeddings.
3. For the synthetic pre-processed matrices, we projected their cells to the same 30-dimensional PC space by left multiplying the 30-by-$p$ loading matrix (with rows as the top 30 loading vectors; $p$ is the number of peaks). Then we projected the synthetic cells from the PC space to the same UMAP space using the `predict()` function in the R package umap. We labeled the cells with their cell types. We reported the miLISI (proposed by Korsunsky et al.[44]) to quantify the degree of mixing between the synthetic cells and real cells (see the subsection "Evaluation metrics"). The miLISI value (ranging from 1 to 2) will be close to 2 if the synthetic cells and real cells are perfectly mixed.

Third, we visualized how scReadSim's and SCAN-ATAC-Sim's synthetic cells mix with real cells in the 2D UMAP embeddings (Supplementary Fig. 6b) using the following steps.

1. We combined the real cells and scReadSim's synthetic cells by concatenating the real peak-by-cell pre-processed matrix and scReadSim's synthetic peak-by-cell pre-processed matrix (from TF-IDF followed by the logarithmic transformation) horizontally as a peak-by-mixed-cell matrix. Similarly, we combined the real cells and SCAN-ATAC-Sim's synthetic cells into a concatenated peak-by-mixed-cell matrix.
2. We projected the two groups of mixed cells to the previously used 30-dimensional PC space by left multiplying the 30-by-$p$ loading matrix (with rows as the top 30 loading vectors; $p$ is the number of peaks) obtained previously from the real peak-by-cell pre-processed matrix.
3. We projected the two groups of mixed cells from the PC space to the previously used UMAP space using the `predict()` function in the R package umap.

**Evaluation metrics.**
- Reads Per Kilobase Million (RPKM). The RPKM value of a feature $i$ is defined as

$$RPKM = \frac{n_i \times 1{,}000{,}000}{N \times l_i}, \tag{1}$$

  where $n_i$ represents the read count of feature $i$, $N = \sum_i n_i$ represents the total read count, and $l_i$ is the length of feature $i$ in kilobase.
- miLISI. The integration local inverse Simpson's Index (iLISI)[44] for each cell measures the effective number of datasets in a neighborhood of the cell. A cell surrounded by neighboring cells from a single dataset has an iLISI value equal to 1; otherwise, if the cell has a neighborhood with an equal number of cells from two datasets, the iLISI value is 2, indicating a perfect mixing of the two datasets in the cell's neighborhood. Hence, the median iLISI (miLISI) of all cells measures the median mixing level of the two datasets across all cells' neighborhoods, and a miLISI close to 2 means that the two datasets have a close-to-perfect mixing.

**Reporting summary**

Further information on research design is available in the Nature Portfolio Reporting Summary linked to this article.

## Data availability

The 10x Genomics single-cell multiome dataset includes ATAC and gene expression data from the embryonic mouse brain tissue[21]. The raw BAM file (ATAC-seq modality) was downloaded from https://cf.10xgenomics.com/samples/cell-arc/2.0.0/e18_mouse_brain_fresh_5k/e18_mouse_brain_fresh_5k_atac_possorted_bam.bam. The raw BAM file (RNA-seq modality) was downloaded from https://s3-us-west-2.amazonaws.com/10x.files/samples/cell-arc/2.0.0/e18_mouse_brain_fresh_5k/e18_mouse_brain_fresh_5k_gex_possorted_bam.bam. The reference genome file (assembly version GRCm38.p4 release M10) was downloaded from GENCODE (https://www.gencodegenes.org/mouse/release_M10.html). The sci-ATAC-seq dataset measures the chromatin accessibility in 17 samples spanning 13 tissues in 8-week-old mice[23]. The raw BAM file was downloaded from http://krishna.gs.washington.edu/content/members/mouse_ATAC_atlas_website/bams/BoneMarrow_62016.bam. Data from the tissue "Bone Marrow 62016" were used for analysis. The reference genome file (assembly version NCBIM37 release M1) was downloaded from GENCODE (https://www.gencodegenes.org/mouse/release_M1.html).

## Code availability

The scReadSim Python package is available at https://github.com/JSB-UCLA/scReadSim. The comprehensive tutorials are available at http://screadsim.readthedocs.io/. In the tutorials, we described the input and output formats, model parameters, and exemplary datasets for each functionality of scReadSim. The source code for reproducing the results are available at: https://github.com/Dominic7227/scReadSim_source[49].

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

## Acknowledgements

The authors appreciate the comments and feedback from Qingyang Wang, Xinzhou Ge, Zhiqian Zhai, and other members of the Junction of Statistics and Biology at UCLA (http://jsb.ucla.edu). The authors thank Dr. Chongzhi Zang for his helpful suggestions. This work was supported by the following grants: National Science Foundation DBI-1846216 and DMS-2113754, NIH/NIGMS R01GM120507 and R35GM140888, Johnson & Johnson WiSTEM2D Award, Sloan Research Fellowship, UCLA David Geffen School of Medicine W.M. Keck Foundation Junior Faculty Award, and Chan-Zuckerberg Initiative Single-Cell Biology Data Insights [Silicon Valley Community Foundation Grant Number: 2022-249355] (to J.J.L.). J.J.L. was a fellow at the Radcliffe Institute for Advanced Study at Harvard University in 2022-2023 while she was writing this paper.

## Author contributions

J.J.L. conceived of the study. G.Y. performed data analysis. G.Y. developed the scReadSim Python package with assistance from D.S. G.Y. and J.J.L. wrote the manuscript.

## Competing interests

The authors declare no competing interests.
