## [Peer Review File · Nature Communications]

scReadSim: a single-cell RNA-seq and ATAC-seq read simulatorREVIEWER COMMENTS

Reviewer #1 (Remarks to the Author):

In this paper, Guanao etc., presented a novel simulator, "scReadSim," for single-cell RNA-seq and single-cell ATAC-seq data. Compared with existing tools in the field, it has several significant advantages. First, it can simulate single-cell RNA-seq and single-cell ATAC-seq, making it the first simulation tool to generate simulated multi-omics data. Second, scReadSim can simulate fastq files to enable benchmarking the entire analysis pipeline, while most of the other simulators only provide count matrices. Lastly, it takes real single-cell data as input to preserve the read sequences, read counts, and read coverage over the whole genome from the real dataset while simulating different sequencing depths and numbers of cells. I envision that researchers will welcome such a useful tool for developing their bioinformatics tools for scRNA-seq and scATAC-seq data. However, I have some comments regarding the functions of the tool.

1. Can authors discuss one of the critical artifacts in single-cell data -- the doublets or multiplets effect? The doublets/multiplets will interfere with the downstream analyses, such as clustering, annotation, and differential analyses, by introducing combined profiles originating from multiple cells. The artifacts exist in most of the 'real' single-cell datasets. So the question is, will the doublets effect be kept or go away after using such data to simulate synthetic datasets?
2. Authors used 10x mouse multi-omics data to demonstrate the simulation on scRNA-seq and scATAC-seq separately. But can the simulated data from the two modalities be integrated so that the scReadSim can be potentially used for benchmarking data integration tools?
3. When comparing the workflow between simulation on scRNA-seq and scATAC-seq (as in Fig1), we can see that the difference is that for scRNA-seq, we can have a known and fixed gene annotation; however, for scATAC-seq, we rely on either the user-inputted trustworthy peaks and non-peak definition or call the peaks from the 'real' data itself. Therefore, for scATAC-seq simulation, the features (peaks in scATAC-seq) are determined by the peak calling process, and the process depends on the algorithm and its parameter settings. Is it possible to start with a superset of possible open chromatin regions, such as the ENCODE cCRE (Candidate Cis-Regulatory Elements) collection summarized from various DNA accessibility assays and ChIP experiments?

Reviewer #2 (Remarks to the Author):

In this manuscript, the authors describe scReadSim, a software tool for generating simulated sequencing reads mimicking scRNA-seq and scATAC-seq experiments. scReadSim generates reads that more closely resemble realistic reads than existing tools, as it effectively samples e.g. read numbers and genomic locations from the real data set used as input. This is certainly a research topic that is relevant to the single-cell community. Overall I find the approach sensible, and the generated data appears realistic. I have a couple of comments on the method itself, and a couple related to the documentation.

1) If I understand correctly, RNA-seq reads are generated as contiguous regions of length l from the genome sequence, starting from the inferred 5' end; i.e., without considering exon boundaries or spliced alignments. Of course, accounting for such effect would make the simulation more complex, and one would have to make a decision e.g. of which isoform to sample from; however, I'm wondering if the current approach has negative downstream effects for quantification, especially for tools that explicitly check whether the reads are compatible with the annotated transcript isoforms, or when separately quantifying exonic/spliced and intronic/unspliced abundances. I think a deeper discussion/investigation of this potential issue would be relevant.

2) Many benchmarking tasks require simulation of data from multiple samples or conditions, with a well-defined ground truth. I was wondering how the authors envision this to be done within the scReadSim framework - would the user provide a separate bam file for each replicate sample, or would they induce the effects artificially in the simulated UMI count matrix? Specifically for the ATAC-seq simulation, if the user does provide separate bam files for each replicate, is there a need for consolidation of peaks between the samples in order to generate an interpretable ground truth for subsequent benchmarking?

3) The documentation is reasonably extensive. However, I found it not completely straightforward to follow the tutorial as it jumps between python and bash commands without explicit mention, and some commands assume that the example data is present in the current working directory (i.e., it assumes that one is in the data/ folder of the installed package). I could also not find the 10X_RNA_chr1_3073253_4526737_unprocess.bam file, which is mentioned in the tutorial, in this folder. I think it would be helpful if the manual was a bit more clear on these points.

4) I downloaded and installed the software and dependencies, and attempted to run the example code provided in https://screadsim.readthedocs.io/en/latest/scRNAseq_10X.html. However, when constructing the count matrix for the genes with `Utility.scRNA_bam2countmat_parallel()`, it failed with the error 'NameError: name 'cells_n' is not defined'.

5) How does scReadSim define 'gene' and 'inter-gene' regions in the presence of overlapping genes? And how does this propagate to the estimation of the counts and the generation of the synthetic count matrix?

6) Cell barcodes are described to be generated randomly. Is there a possibility to provide a list of 'allowed' barcodes - e.g. to avoid simulated reads being filtered out by downstream tools since they don't conform to a pre-defined list of possible barcodes?

7) For the evaluation of the UMI deduplication tools (Fig S13), I'm not sure that miLISI is the most suitable metric, as it just checks whether the two data sets overlap, not whether it's indeed the corresponding cells that end up close to each other. Perhaps additionally checking whether the corresponding 'ground truth' cell is in fact among the nearest neighbors of a quantified cell would be useful.

Response to reviewers' comments on “scReadSim: a single-cell RNA-seq and ATAC-seq read simulator”

We thank the two reviewers for their constructive comments. We have addressed all their comments and modified the paper accordingly. Our detailed point-by-point answers are on the following pages.

Please note that reviewers' comments are in **blue**, our answers are in **black**, and quotes from our revised manuscript are in **brown**.

Changes are indicated in **blue** in the revised manuscript.

Answers to Reviewer 1

In this paper, Guanao et al., presented a novel simulator, “scReadSim,” for single-cell RNA-seq and single-cell ATAC-seq data. Compared with existing tools in the field, it has several significant advantages. First, it can simulate single-cell RNA-seq and single-cell ATAC-seq, making it the first simulation tool to generate simulated multi-omics data. Second, scReadSim can simulate fastq files to enable benchmarking the entire analysis pipeline, while most of the other simulators only provide count matrices. Lastly, it takes real single-cell data as input to preserve the read sequences, read counts, and read coverage over the whole genome from the real dataset while simulating different sequencing depths and numbers of cells. I envision that researchers will welcome such a useful tool for developing their bioinformatics tools for scRNA-seq and scATAC-seq data. However, I have some comments regarding the functions of the tool.

We thank the reviewer for appreciating the merit of our work.

Comment R1.1 Can authors discuss one of the critical artifacts in single-cell data – the doublets or multiplets effect? The doublets/multiplets will interfere with the downstream analyses, such as clustering, annotation, and differential analyses, by introducing combined profiles originating from multiple cells. The artifacts exist in most of the ‘real’ single-cell datasets. So the question is, will the doublets effect be kept or go away after using such data to simulate synthetic datasets?

Answer to R1.1 We thank the reviewer for this insightful question. We totally agree that the existence of doublets/multiplets in single-cell data would affect downstream analysis. We would like to clarify that, as scReadSim mimics real data, it would require doublets/multiplets to be removed from real data so its synthetic data would not contain doublets/multiplets. Accordingly, we have updated scReadSim by adding a step to identify and remove doublets/multiplets from real data.

To explain the newly added doublet/multiplet-removal step in scReadSim, we have added **Section “Doublet/multiplet detection”** to **Methods** (see below) and updated our software tutorial.

“In single-cell sequencing technologies that use droplets or wells (called reaction volumes) to isolate single cells, doublets/multiplets exist when more than one cell is captured by the reaction volume and mistaken as a single cell. Doublets/multiplets would bias downstream data analysis [3, 6, 7]. As scReadSim mimics real data, it requires doublets/multiplets to be removed from real data so its synthetic data would not contain doublets/multiplets. To remove doublets/multiplets from real data, scReadSim uses scDbfFinder [2] as the doublet detection method based on the benchmark study [7]. For scRNA-seq data, scDbfFinder is applied to the real gene-by-cell UMI count matrix to identify the columns (“cells”) that likely correspond to doublets/multiplets; then,

the identified columns would be removed from both the real gene-by-cell and inter-gene-by-cell UMI count matrix. For scATAC-seq data, scDbfFinder is applied to the real peak-by-cell read count matrix to identify the columns (“cells”) that likely correspond to doublets/multiplets; then, the identified columns would be removed from the real peak-by-cell and non-peak-by-cell read count matrices. Finally, after removing the doublets/multiplets from real data, scReadSim is trained on the post-removal real count matrices to generate synthetic data.”

Comment R1.2 Authors used 10x mouse multi-omics data to demonstrate the simulation on scRNA-seq and scATAC-seq separately. But can the simulated data from the two modalities be integrated so that the scReadSim can be potentially used for benchmarking data integration tools?

Answer to R1.2 We thank the reviewer for this constructive suggestion. Indeed, scReadSim is able to generate synthetic reads for single-cell multi-omics data while providing the ground truths for both modalities, including the true UMI counts for scRNA-seq and the open chromatin regions for scATAC-seq. In order to do so, scReadSim requires users to input a single-cell multi-omics (simultaneously measured scRNA-seq and scATAC-seq) dataset. Then, scReadSim processes the scRNA-seq and scATAC-seq modalities to construct real count matrices, including scRNA-seq gene-by-cell and inter-gene-by-cell UMI count matrices and scATAC-seq peak-by-cell and non-peak-by-cell read count matrices. Given these multi-omic feature-by-cell count matrices, scReadSim implements scDesign3 [5], our recently developed single-cell multi-omics count simulator, to simulate synthetic cells’ UMI and read counts for both modalities. Finally, based on the synthetic counts, scReadSim generates synthetic reads for both modalities.

To demonstrate how to use scReadSim to simulate single-cell multi-omics data, we have added the following **Section “scReadSim for single-cell multi-omics”** to **Methods** (see below). For users’ convenience, we have also provided a tutorial on the scReadSim software webpage: <https://screadsim.readthedocs.io/en/latest/scMultiOmics.html>.

“The multi-modality of scReadSim enables it to simulate single-cell multi-omics (simultaneously measured scRNA-seq and scATAC-seq modalities) sequencing reads. In order to do so, scReadSim requires users to input a single-cell multi-omics dataset and the corresponding reference genome. Following the procedures for simulating individual scRNA-seq and scATAC-seq modalities described above, scReadSim first processes the two modalities to construct real count matrices, including scRNA-seq gene-by-cell and inter-gene-by-cell UMI count matrices and scATAC-seq peak-by-cell and non-peak-by-cell read count matrices. Then, given these count matrices, scReadSim implements scDesign3 [5], a single-cell multi-omics count simulator, to simulate synthetic cells’ UMI and read counts for both modalities. Finally, based on the synthetic counts, scReadSim generates synthetic reads for both modalities in the FASTQ format, along with a list of synthetic

cell barcodes. As a result, the synthetic single-cell multi-omics sequencing reads would contain synthetic cells with both modalities and ground truths, including open chromatin regions for the scATAC-seq modality and UMI abundance for the scRNA-seq modality.”

Regarding the reviewer’s question about whether the synthetic multi-omics data generated by scReadSim can help benchmark data integration tools, we would like to clarify that the existing data integration tools work for count data only, not reads. Hence, benchmarking these tools only requires single-cell multi-omics count data, which scDesign3 can generate. Although scReadSim is able to provide synthetic multi-omics reads with ground truths, they need to be summarized into count matrices first before data integration tools can be applied.

Comment R1.3 When comparing the workflow between simulation on scRNA-seq and scATAC-seq (as in Fig1), we can see that the difference is that for scRNA-seq, we can have a known and fixed gene annotation; however, for scATAC-seq, we rely on either the user-inputted trustworthy peaks and non-peak definition or call the peaks from the ‘real’ data itself. Therefore, for scATAC-seq simulation, the features (peaks in scATAC-seq) are determined by the peak calling process, and the process depends on the algorithm and its parameter settings. Is it possible to start with a superset of possible open chromatin regions, such as the ENCODE cCRE (Candidate Cis-Regulatory Elements) collection summarized from various DNA accessibility assays and ChIP experiments?

Answer to R1.3 We thank the reviewer for this insightful suggestion. Our answer to the reviewer’s question is yes, scReadSim can take pre-defined possible open chromatin regions (such as the ENCODE cCREs) as input for scATAC-seq read simulation, just as scReadSim uses pre-defined genes for scRNA-seq read simulation.

Accordingly, we have added an alternative option for scReadSim to allow users to input a pre-defined set of possible peaks. Given the input set, scReadSim determines trustworthy peaks as the input peaks that overlap with the real-data peaks identified by MACS3 under a stringent criterion ($q\text{-value} = 0.01$). Among the regions complementary to the input peaks, scReadSim determines trustworthy non-peaks as the regions that overlap with the real-data non-peaks identified by MACS3 under a stringent criterion (non-peaks are complementary to the peaks identified at $q\text{-value} = 0.1$). This refinement step that defines trustworthy peaks and non-peaks is allow scReadSim to provide ground-truth peaks and mimic real data simultaneously.

To explain how scReadSim handles user-input possible peaks, we have added the following paragraph to **Section “scReadSim for scATAC-seq”** in **Methods** and updated our software tutorial.

“To pre-process real scATAC-seq data for training, scReadSim requires a BAM file (containing scATAC-seq reads in cells) and users’ trustworthy peaks and non-peaks in the input BAM file.

Alternatively, if users do not specify trustworthy peaks and non-peaks, scReadSim provides users with two options for generating trustworthy peaks and non-peaks. Option One is that scReadSim first deploys the peak-calling tool MACS3 [8] to identify the trustworthy peaks using a stringent rule (by setting a q-value threshold 0.01) and the trustworthy non-peaks using a stringent rule (the non-peaks are complementary to the peaks called under the q-value threshold 0.1). Option Two allows users to input a set of possible peaks from public databases, for instance, the ENCODE cCREs (candidate cis-regulatory elements) [4] summarized from various DNA accessibility assays and ChIP-seq experiments. This input set of possible peaks does not depend on the input BAM file, similar to the input gene annotation scReadSim uses to generate scRNA-seq reads. To define trustworthy peaks and non-peaks under this Option Two, scReadSim selects the input possible peaks and non-peaks (defined as the regions complementary to the input possible peaks) that overlap with the identified peaks and non-peaks by MACS3 under Option One. The definition of overlapping is if an input possible peak (or non-peak) has half of its length covered by one or more MACS3-identified peaks (or non-peaks); then such an input possible peak (or non-peak) would be considered trustworthy.”

Answers to Reviewer 2

In this manuscript, the authors describe scReadSim, a software tool for generating simulated sequencing reads mimicking scRNA-seq and scATAC-seq experiments. scReadSim generates reads that more closely resemble realistic reads than existing tools, as it effectively samples e.g. read numbers and genomic locations from the real data set used as input. This is certainly a research topic that is relevant to the single-cell community. Overall I find the approach sensible, and the generated data appears realistic. I have a couple of comments on the method itself, and a couple related to the documentation.

We thank the reviewer for summarizing our manuscript and appreciating the merit of our work.

Comment R2.1 If I understand correctly, RNA-seq reads are generated as contiguous regions of length l from the genome sequence, starting from the inferred 5' end; i.e., without considering exon boundaries or spliced alignments. Of course, accounting for such effect would make the simulation more complex, and one would have to make a decision e.g. of which isoform to sample from; however, I'm wondering if the current approach has negative downstream effects for quantification, especially for tools that explicitly check whether the reads are compatible with the annotated transcript isoforms, or when separately quantifying exonic/spliced and intronic/unspliced abundances. I think a deeper discussion/investigation of this potential issue would be relevant.

Answer to R2.1 We thank the reviewer for the insightful comment and constructive suggestion. We acknowledge that the current scReadSim only provides gene-level ground truths (UMI counts) for scRNA-seq read generation, and it does not yet have the functionality to provide isoform-level ground truths (including true isoforms and their abundance). Hence, we do not recommend users to use the current version of scReadSim to benchmark single-cell computational tools that study isoforms or alternative splicing events. As a future direction, we will extend scReadSim to the isoform level by generating synthetic scRNA-seq reads with isoform-level ground truths while mimicking real data.

To avoid possible confusion, we have added the following discussion to **Discussion**.

“Note that the current version of scReadSim generates scRNA-seq reads with gene-level ground truths (UMI counts) only, not isoform-level ground truths including isoform abundance and alternative splicing events. Therefore, we do not recommend using the current version of scReadSim to benchmark computational tools that study isoforms or alternative splicing events. As a future direction, we will generalize scReadSim to generate scRNA-seq short and long reads with ground-truth RNA isoform abundance, a functionality unavailable in existing scRNA-seq simulators.”

Comment R2.2 Many benchmarking tasks require simulation of data from multiple samples or conditions, with a well-defined ground truth. I was wondering how the authors envision this to be done within the scReadSim framework — would the user provide a separate bam file for each replicate sample, or would they induce the effects artificially in the simulated UMI count matrix? Specifically for the ATAC-seq simulation, if the user does provide separate bam files for each replicate, is there a need for consolidation of peaks between the samples in order to generate an interpretable ground truth for subsequent benchmarking?

Answer to R2.2 We thank the reviewer for the constructive questions. We agree that adding a multiple-sample/condition module is critical for scReadSim. Hence, we have updated scReadSim with a new module to generate synthetic scRNA-seq and scATAC-seq reads from multiple samples or conditions. The new module requires users to specify a separate BAM file for each sample/condition.

As a future direction, we would like to generate multiple samples/conditions from one input BAM file by artificially introducing sample/condition effects. The challenge here is to make the artificially introduced effects realistic. Toward this goal, we will leverage our newly developed count simulator scDesign3 that allows for adding sample/condition effects. We have added the following discussion to **Discussion**.

“To generate multiple samples and conditions, the current version of scReadSim requires users to input one BAM file per sample or condition. As a future direction, we will leverage the count simulator scDesign3 [5], which allows for adding sample and condition effects, to generate synthetic reads of multiple samples and conditions when the input is only one BAM file, by arbitrarily introducing sample and condition effects from realistic distributions.”

To answer the reviewer’s last question about the specification of ground-truth peaks when multiple scATAC-seq data are inputted, we have implemented the following in our newly added multi-sample/condition module. Thanks to **Reviewer 1’s Comment R1.3**, in this new module, we let scReadSim start from a pre-specified input list of possible peaks, for instance, the ENCODE cCREs (Candidate Cis-Regulatory Elements) [4] defined based on various DNA accessibility and ChIP-seq assays. This input peak list provides a consistent segmentation of the reference genome, allowing scReadSim to define comparable ground-truth peaks across multiple samples. (Please see **Answer to R1.3** for detail.)

We have also added the following **Section “scReadSim for multiple samples/conditions”** to **Methods** to explain how scReadSim simulates synthetic reads under the multi-sample/condition scenario. In addition, we have added two tutorials (RNA-seq: https://screadsim.readthedocs.io/en/latest/scRNAseq_MultiSamples.html; ATAC-seq: https://screadsim.readthedocs.io/en/latest/scATACseq_MultiSamples.html) to the scReadSim software webpage to demonstrate how to use the newly added multi-sample/condition module.

“scReadSim provides a module for users to simulate single-cell sequencing reads from multiple samples or conditions. It requires users to input multiple BAM files corresponding to multiple samples or conditions, as well as the corresponding reference genome file. Then, scReadSim trains a model for each input BAM file and simulates synthetic reads separately for each sample or condition. Specifically for scATAC-seq, to provide comparable ground-truth peaks across the multiple synthetic samples, scReadSim requires users to input a set of possible peaks, such as the ENCODE cCREs (Candidate Cis-Regulatory Elements) [4]. This pre-specified peak set provides scReadSim with a consistent segmentation of the reference genome, and scReadSim’s ground-truth peaks will be defined based on this segmentation. ”

Comment R2.3 The documentation is reasonably extensive. However, I found it not completely straightforward to follow the tutorial as it jumps between python and bash commands without explicit mention, and some commands assume that the example data is present in the current working directory (i.e., it assumes that one is in the data/ folder of the installed package). I could also not find the `10X_RNA_chr1_3073253_4526737_unprocess.bam` file, which is mentioned in the tutorial, in this folder. I think it would be helpful if the manual was a bit more clear on these points.

Answer to R2.3 We thank the reviewer for the detailed check and constructive suggestion. We have significantly restructured our tutorials. In particular, we have made a clear distinction between bash and Python commands; and improved the documentation by distinguishing the mandatory steps from the optional ones. We have tested the commands in our tutorials on multiple Linux servers, and we will continue maintaining our software package and tutorials based on user feedback.

Comment R2.4 I downloaded and installed the software and dependencies, and attempted to run the example code provided in https://screadsim.readthedocs.io/en/latest/scRNAseq_10X.html. However, when constructing the count matrix for the genes with `Utility.scRNA_bam2countmat_parallel()`, it failed with the error `NameError: name 'cells_n' is not defined`.

Answer to R2.4 We apologize for the bug in the previous version of our package. We did not encounter this error with the latest version of scReadSim, so we think this error has been fixed in our package updates. We have performed multiple rounds of independent tests and now distributed the latest version of scReadSim on both PyPI and GitHub.

Comment R2.5 How does scReadSim define ‘gene’ and ‘inter-gene’ regions in the presence of overlapping genes? And how does this propagate to the estimation of the counts and the generation

of the synthetic count matrix?

Answer to R2.5 We thank the reviewer for this clarification question. When preparing features for scRNA-seq data, scReadSim takes the union region of overlapping genes if they exist and replaces these overlapping genes with one supergene. Then, non-overlapping genes and supergenes are used to construct the gene-by-cell UMI count matrix and also used for generating the synthetic count matrix.

To avoid possible confusion for readers, we have added the following sentences to the **Section “scReadSim for scRNA-seq”** of **Methods** in the revised manuscript.

“Specifically, to account for the overlapping genes in the annotation file, scReadSim first aggregates overlapping genes into non-overlapping regions (called “supergenes”) to avoid double counting of reads from overlapping genes. Then scReadSim refers to non-overlapping genes and supergenes as “genes” and defines inter-genes as the complementary regions. We acknowledge that this simple strategy, though avoiding double counting of reads, may obscure the read coverage differences of overlapping genes. As a future direction, by extending scReadSim to an isoform-level read simulator, we can solve this problem by providing the ground-truth isoforms and their abundance.”

Comment R2.6 Cell barcodes are described to be generated randomly. Is there a possibility to provide a list of ‘allowed’ barcodes – e.g. to avoid simulated reads being filtered out by downstream tools since they don’t conform to a pre-defined list of possible barcodes?

Answer to R2.6 We thank the reviewer for this expert suggestion. We would like to clarify that scReadSim does generate a list of synthetic cell barcodes while outputting the synthetic reads in FASTQ files. This cell barcode list allows users to perform downstream analysis with our synthetic reads. Thanks to the reviewer’s comment, we have now stated clearly the output of scReadSim by adding the following sentence to the **Section “Output in FASTQ or BAM formats”** of **Methods**. We have also updated our tutorials on the scReadSim software webpage <http://screadsim.readthedocs.io/> to list the output of each step. We hope this added information can help users better understand how to implement scReadSim and utilize synthetic data.

“In addition, the list of synthetic cell barcodes would be outputted for users’ reference, which could serve as a cell barcode white list for downstream analyses.”

Comment R2.7 For the evaluation of the UMI deduplication tools (Fig S13), I’m not sure that miLISI is the most suitable metric, as it just checks whether the two data sets overlap, not whether it’s indeed the corresponding cells that end up close to each other. Perhaps additionally checking whether the corresponding ‘ground truth’ cell is in fact among the nearest neighbors of a quantified

cell would be useful.

Answer to R2.7 We thank the reviewer for this insightful suggestion and agree that miLISI measures the mixing degree of two sets of cells but misses the one-to-one cell correspondences. We have replaced miLISI with the Euclidean distance between a synthetic cell's two UMAP coordinates, calculated based on the ground-truth UMI counts and the deduplicated UMI counts, respectively. The mean value of the Euclidean distances of all synthetic cells is displayed for each UMI deduplication tool in Figs. R1 and R2 (corresponding to **Fig. 2g** in **Main Text** and **Fig. S13** in **Supplementary Figures**). A smaller value indicates that the deduplicated UMI count matrix better agrees with the ground truth UMI count matrix in UMAP visualization.

Figure R1: (Caption on next page.)

Figure R1: **a–b**, scReadSim’s synthetic scATAC-seq data mimic the real sci-ATAC-seq dataset [1] in terms of the cell-type-specific read coverage (**a**), the fragment-size distribution (**b** Top), substitution error rate per base call within a read (**b** Middle), and the peaks called by MACS3 at the pseudo-bulk level (**b** Bottom; **Methods**). **c**, scReadSim enables user-designed, cell-type-specific ground-truth (g.t.) peaks for scATAC-seq read generation. **d–g**, Benchmark of UMI deduplication tools using scReadSim’s synthetic scRNA-seq reads in four aspects: **d**, Time usage of deduplication tools on synthetic datasets with varying cell numbers (at a fixed sequencing depth) or varying sequencing depths (at a fixed cell number). The y-axis indicates the time lapse (in seconds), and the x-axis shows the number of synthetic cells (left) or the total number of UMIs (sequencing depth, right). **e**, Distributions of summary statistics of the UMI count matrices (ground truth, cellranger’s output, and UMI-tools’ output) at the gene level (mean, variance, coefficient of variance (cv), and zero proportion) and the cell level (zero proportion and library size). **f**, Cell-wise and gene-wise correlations (Pearson correlation and Kendell’s tau) between the ground-truth UMI count matrix and each deduplication tool’s output UMI count matrix. **g**, UMAP visualizations of the ground-truth UMI count matrix and each deduplication tool’s output UMI count matrix. Cells are colored by the cell clusters outputted by scReadSim (the clusters are from the real data used to train scReadSim). The synthetic cells’ deduplicated UMI counts are projected to the UMAP space defined by the same cells’ ground-truth UMI counts. We computed the Euclidean distance between each synthetic cell’s two UMAP coordinates, calculated based on the cell’s ground-truth UMI counts and deduplicated UMI counts, respectively. The mean value of the Euclidean distances of all synthetic cells is displayed for each UMI deduplication tool: a smaller value indicates that the deduplicated UMI count matrix better agrees with the ground truth UMI count matrix in UMAP visualization. **h–i**, Benchmark of peak calling tools using scReadSim’s synthetic scATAC-seq reads through in two aspects: **h**, Distributions of RPKM values of peak and non-peak regions in the ground truth (specified in scReadSim) and each tool’s peak-calling result. Mean differences (mean diff.) of peak regions’ RPKM are calculated between ground truth and each method. **i**, true positive rate (TPR) vs. false positive rate (FPR) curves (top) and precision vs. recall curves (bottom) using user-designed open chromatin regions as the ground-truth peaks. A called peak is considered true if it overlaps at least 225 bp of a user-designed open chromatin region, where 225 bp is half of the shortest open chromatin region.

Figure R2: UMAP visualizations of the ground-truth UMI count matrix and each deduplication tool’s output UMI count matrix. Cells are colored by the cell clusters outputted by scReadSim (the clusters are from the real data used to train scReadSim). The synthetic cells’ deduplicated UMI counts are projected to the UMAP space defined by the same cells’ ground-truth UMI counts. We computed the Euclidean distance between each synthetic cell’s two UMAP coordinates, calculated based on the cell’s ground-truth UMI counts and deduplicated UMI counts, respectively. The mean value of the Euclidean distances of all synthetic cells is displayed for each UMI deduplication tool: a smaller value indicates that the deduplicated UMI count matrix better agrees with the ground truth UMI count matrix in UMAP visualization.

References

- [1] Cusanovich, D. A., Hill, A. J., Aghamirzaie, D., Daza, R. M., Pliner, H. A., Berletch, J. B., Filippova, G. N., Huang, X., Christiansen, L., DeWitt, W. S., et al. (2018). A single-cell atlas of in vivo mammalian chromatin accessibility. *Cell*, 174(5):1309–1324.
- [2] Germain, P.-L., Lun, A., Meixide, C. G., Macnair, W., and Robinson, M. D. (2021). Doublet identification in single-cell sequencing data using scdblfinder. *F1000Research*, 10.
- [3] Germain, P.-L., Sonrel, A., and Robinson, M. D. (2020). pipecomp, a general framework for the evaluation of computational pipelines, reveals performant single cell rna-seq preprocessing tools. *Genome biology*, 21(1):1–28.
- [4] Moore, J. E., Purcaro, M. J., Pratt, H. E., Epstein, C. B., Shores, N., Adrian, J., Kawli, T., Davis, C. A., Dobin, A., et al. (2020). Expanded encyclopaedias of dna elements in the human and mouse genomes. *Nature*, 583(7818):699–710.
- [5] Song, D., Wang, Q., Yan, G., Liu, T., Sun, T., and Li, J. J. (2023). scdesign3 generates realistic in silico data for multimodal single-cell and spatial omics. *Nature Biotechnology*, pages 1–6.
- [6] Xi, N. M. and Li, J. J. (2021a). Benchmarking computational doublet-detection methods for single-cell rna sequencing data. *Cell systems*, 12(2):176–194.
- [7] Xi, N. M. and Li, J. J. (2021b). Protocol for executing and benchmarking eight computational doublet-detection methods in single-cell rna sequencing data analysis. *STAR protocols*, 2(3):100699.
- [8] Zhang, Y., Liu, T., Meyer, C. A., Eeckhoute, J., Johnson, D. S., Bernstein, B. E., Nusbaum, C., Myers, R. M., Brown, M., Li, W., et al. (2008). Model-based analysis of chip-seq (macs). *Genome biology*, 9(9):1–9.

REVIEWER COMMENTS

Reviewer #1 (Remarks to the Author):

Thank the authors for addressing all my comments and adding new content to the manuscript and new features to the tool! I don't have further comments.

Reviewer #2 (Remarks to the Author):

The authors have nicely addressed most of my concerns by expanding the manuscript and documentation and extending the capabilities of the software.

Also with version 1.4.0, I got the error "NameError: name 'cells_n' is not defined" when running `Utility.scRNA_bam2countmat_parallel()` following the code in the tutorial. However, disabling the parallelization by setting `"n_cores=1"` solved the problem and I was able to run through the tutorial (I'm on an M1 mac, not sure whether that is the reason). I don't consider this a showstopper, but I would suggest (if possible) adding some additional checks in the code to detect issues as early as possible and provide a more informative error message.

Regarding my first comment (R2.1), I think the current way of generating RNA-seq reads as contiguous regions of the genome can be an issue also when only gene-level analyses are of interest. For example, STARsolo checks whether reads fall completely within annotated exons, even if the goal is gene-level quantification [1]. Thus, a read that extends into an intron (which it may do since the transcript structure is not taken into account when simulating) may not be counted towards the gene in the default counting mode. CellRanger currently counts also intronic reads - could that have an influence on the benchmarking of UMI deduplication tools performed in the manuscript?

[1] This section is from the STARsolo preprint (<https://www.biorxiv.org/content/10.1101/2021.05.05.442755v1>), section 5.1.2:

"For each read alignment, we find a set of concordant annotated transcripts. The strandedness of the scRNA-seq library is user-specified with the `--soloStrand` option. An alignment is considered concordant with a transcript if all of its alignment blocks are contained within transcript exons. If splice junctions are present in the alignment, they have to agree with the transcript junctions. From the set of concordant transcripts we create a set of corresponding concordant genes."

Response to reviewers' comments on “scReadSim: a single-cell RNA-seq and ATAC-seq read simulator”

We thank Reviewer 2 for the further, constructive comments. We have addressed the comments and modified the paper accordingly. Our detailed point-by-point answers are on the following pages.

Please note that reviewers' comments are in **blue**, our answers are in **black**, and quotes from our revised manuscript are in **brown**.

Changes are indicated in **blue** in the revised manuscript.

Answers to Reviewer 1

Thank the authors for addressing all my comments and adding new content to the manuscript and new features to the tool! I don't have further comments.

We thank the reviewer for providing thoughtful comments for us to improve our manuscript.

Answers to Reviewer 2

The authors have nicely addressed most of my concerns by expanding the manuscript and documentation and extending the capabilities of the software.

We thank the reviewer for the constructive comments.

Comment R2.1 Also with version 1.4.0, I got the error `NameError: name 'cells_n' is not defined` when running `Utility.scRNA_bam2countmat_parallel()`, following the code in the tutorial. However, disabling the parallelization by setting `n_cores=1` solved the problem and I was able to run through the tutorial (I'm on an M1 mac, not sure whether that is the reason). I don't consider this a showstopper, but I would suggest (if possible) adding some additional checks in the code to detect issues as early as possible and provide a more informative error message.

Answer to R2.1 We thank the reviewer for the detailed check and constructive suggestion. We have corrected the bug and updated our functions by adding sanity-check steps.

Comment R2.2 Regarding my first comment (R2.1), I think the current way of generating RNA-seq reads as contiguous regions of the genome can be an issue also when only gene-level analyses are of interest. For example, STARsolo checks whether reads fall completely within annotated exons, even if the goal is gene-level quantification [1]. Thus, a read that extends into an intron (which it may do since the transcript structure is not taken into account when simulating) may not be counted towards the gene in the default counting mode. CellRanger currently counts also intronic reads - could that have an influence on the benchmarking of UMI deduplication tools performed in the manuscript?

[1] This section is from the STARsolo preprint (<https://www.biorxiv.org/content/10.1101/2021.05.05.442755v1>), section 5.1.2:

“For each read alignment, we find a set of concordant annotated transcripts. The strandedness of the scRNA-seq library is user-specified with the `soloStrand` option. An alignment is considered concordant with a transcript if all of its alignment blocks are contained within transcript exons. If splice junctions are present in the alignment, they have to agree with the transcript junctions. From the set of concordant transcripts we create a set of corresponding concordant genes.”

Answer to R2.2 We thank the reviewer for this expert suggestion. The previous version of scReadSim indeed generates intronic synthetic reads that may affect the performance of cellranger (v7.0.0, counting intronic reads by default). To account for the potential bias induced by these intronic reads, we re-performed our benchmark study for UMI deduplication tools by counting exonic reads only. Specifically, we implemented cellranger (v7.0.0) by setting the flag `--include-introns`

to false. We further included STARsolo in our benchmark study to compare with UMI-tools, cellranger, and Alevin. Our latest benchmark results indicate that UMI-tools achieves the best accuracy, while Alevin and STARsolo are more computationally efficient. Accordingly, we have updated our benchmark results in **Results Section “Application 1: Benchmarking UMI deduplication tools.”**

“For UMI-based scRNA-seq data, UMI deduplication tools were developed to quantify gene expression levels from scRNA-seq reads, some of which may come from the same RNA molecule. UMI deduplication tools input scRNA-seq reads (containing UMIs and cell barcodes) and output a gene-by-cell UMI count matrix. scReadSim can be used for benchmarking UMI deduplication tools because it provides the ground-truth UMI count matrix in its synthetic scRNA-seq read generation process. Hence, we deployed scReadSim to the mouse 10x single-cell Multiome dataset [1] (the RNA-seq modality only) and used the synthetic reads to benchmark popular UMI deduplication tools: cellranger [9], STARsolo [4], UMI-tools [7], and Alevin [8]. Since Alevin is a transcript-level deduplication tool but scReadSim generates gene-level scRNA-seq reads, we will focus on cellranger, STARsolo, and UMI-tools in the following result comparison (Fig. R1d–g); the Alevin result are in Figs. R2 and R3. Our benchmark study shows that UMI-tools achieves better accuracy than STARsolo and cellranger, while STARsolo is most computationally efficient among the three tools.

First, in terms of computational efficiency, STARsolo runs faster than UMI-tools and cellranger on all synthetic datasets (with varying cell numbers and sequencing depths) (Fig. R1d). As expected, all three tools take longer to run when the sequencing depth increases, and their running time is unaffected by the cell number when the sequencing depth is fixed.

Second, the UMI count matrix output by UMI-tools agrees better with the ground-truth UMI count matrix in terms of (1) the distributions of six summary statistics, including four gene-level statistics (mean, variance, coefficient of variance, and zero proportion) and two cell-level statistics (zero proportion and cell library size) (Fig. R1e); and (2) Pearson correlations and Kendall’s tau correlations between the two matrices, both gene-wise (the correlation across cells per gene) and cell-wise (the correlation across genes per cell) (Fig. R1f).

Third, UMAP visualization shows that, UMI-tools outputs a UMI count matrix that is most similar to the ground-truth UMI count matrix (evidenced by the smallest average Euclidean distance between synthetic cells’ two UMAP coordinates, calculated based on the ground-truth UMI counts and deduplicated UMI counts, respectively) and best preserves the separation among cell clusters (Fig. R1g).”

To further address the potential issue caused by intronic reads of scReadSim, we updated our software by adding an alternative mode to generate exonic reads only: instead of extracting reads

from the genome, scReadSim can generate exonic reads from a collapsed transcriptome (similar to the collapsed transcriptome used in UMI-tools [7] and the superTranscript concept proposed by [3]). Specifically, to generate the collapsed transcriptome, we first took the union of all transcripts into one collapsed transcript for each gene, and we then generated the transcriptome FASTA file using the reference genome and the collapsed transcript annotations for all genes.

To demonstrate how to use scReadSim to simulate scRNA-seq exon-only reads, we have added the following paragraph to **Methods Section “scReadSim for scRNA-seq”** (see below). For users’ convenience, we have also provided a tutorial on the scReadSim software webpage: https://screadsim.readthedocs.io/en/latest/scRNAseq_10X_ExonOnly.html.

“Alternatively, in (4), scReadSim enables users to generate scRNA-seq synthetic reads from the transcriptome. To do so, scReadSim requires a collapsed transcriptome FASTA file. This collapsed transcriptome can be generated from a reference genome and a gene annotation GTF/GFF file by (1) combining all transcripts of a gene into a collapsed transcript with the software `cgat` [6]; (2) generating the transcriptome FASTA file based on the reference genome and the collapsed transcript annotations using the software `gffread` [5]. For each gene, scReadSim uniformly samples the c synthetic reads’ 5’ positions from $[0, L_t]$, where L_t is the gene’s collapsed transcript length; then scReadSim finds the c synthetic reads’ 3’ positions in the collapsed transcript based on the read length. Next, given every synthetic read’s 5’ and 3’ positions in the collapsed transcript, scReadSim extracts the read sequence from the collapsed transcript sequence.”

To clarify scReadSim’s scope and avoid users’ misuse of our software, we have added the following paragraph to **Discussion**. As a future direction, we will extend scReadSim to a transcript-level read simulator, which can completely solve this intronic-read issue and meanwhile provide transcript abundances as higher-resolution ground truths.

“In addition, the default implementation of scReadSim may generate intronic synthetic reads when extracting sequences from the reference genome. Therefore, for benchmarking scenarios where intronic and exonic reads might make a difference, such as implementing `cellranger` (v7.0.0) under the default mode (counting intronic reads), we recommend using the transcriptome mode of scReadSim to generate exonic reads by extracting sequences from a collapsed transcriptome, compiled from genes’ annotated transcripts [3, 7].”

Figure R1: (Caption on next page.)

Figure R1: **a–b**, scReadSim’s synthetic scATAC-seq data mimic the real sci-ATAC-seq dataset [2] in terms of the cell-type-specific read coverage (**a**), the fragment-size distribution (**b** Top), substitution error rate per base call within a read (**b** Middle), and the peaks called by MACS3 at the pseudo-bulk level (**b** Bottom; **Methods**). **c**, scReadSim enables user-designed, cell-type-specific ground-truth (g.t.) peaks for scATAC-seq read generation. **d–g**, Benchmark of UMI deduplication tools using scReadSim’s synthetic scRNA-seq reads in four aspects: **d**, Time usage of deduplication tools on synthetic datasets with varying cell numbers (at a fixed sequencing depth) or varying sequencing depths (at a fixed cell number). The y-axis indicates the time lapse (in seconds), and the x-axis shows the number of synthetic cells (left) or the total number of UMIs (sequencing depth, right). **e**, **Distributions of summary statistics of the UMI count matrices** (ground truth, cellranger’s output, UMI-tools’ output, and STARsolo’s output) at the gene level (mean, variance, coefficient of variance (cv), and zero proportion) and the cell level (zero proportion and library size). **f**, Cell-wise and gene-wise correlations (Pearson correlation and Kendell’s tau) between the ground-truth UMI count matrix and each deduplication tool’s output UMI count matrix. **g**, UMAP visualizations of the ground-truth UMI count matrix and each deduplication tool’s output UMI count matrix. Cells are colored by the cell clusters outputted by scReadSim (the clusters are from the real data used to train scReadSim). The synthetic cells’ deduplicated UMI counts are projected to the UMAP space defined by the same cells’ ground-truth UMI counts. We computed the Euclidean distance between each synthetic cell’s two UMAP coordinates, calculated based on the cell’s ground-truth UMI counts and deduplicated UMI counts, respectively. The mean value of the Euclidean distances of all synthetic cells is displayed for each UMI deduplication tool: a smaller value indicates that the deduplicated UMI count matrix better agrees with the ground truth UMI count matrix in UMAP visualization. **h–i**, Benchmark of peak calling tools using scReadSim’s synthetic scATAC-seq reads through in two aspects: **h**, Distributions of RPKM values of peak and non-peak regions in the ground truth (specified in scReadSim) and each tool’s peak-calling result. Mean differences (mean diff.) of peak regions’ RPKM are calculated between ground truth and each method. **i**, true positive rate (TPR) vs. false positive rate (FPR) curves (top) and precision vs. recall curves (bottom) using user-designed open chromatin regions as the ground-truth peaks. A called peak is considered true if it overlaps at least 225 bp of a user-designed open chromatin region, where 225 bp is half of the shortest open chromatin region.

Figure R2: Benchmark of UMI deduplication tools using scReadSim’s synthetic scRNA-seq reads. The ground-truth UMI count matrix is the gene-by-cell UMI count matrix generated by scReadSim. **Four deduplication tools are considered: cellranger, Alevin, UMI-tools, and STARsolo.** **a**, Time usage of deduplication tools on synthetic datasets with varying cell numbers or sequencing depths. The y-axis indicates the time lapse (in seconds), and the x-axis reflects the number of synthetic cells or the total number of synthetic reads (sequencing depth). **b**, Cell-wise and gene-wise correlations (Pearson correlation and Kendall’s tau) between the ground-truth UMI count matrix and each deduplication tool’s output UMI count matrix. **c**, Summary statistics of the UMI count matrices at the gene level (mean, variance, coefficient of variance (cv), and zero proportion) and the cell level (zero proportion and cell library size).

Figure R3: UMAP visualizations of the ground-truth UMI count matrix and each deduplication tool’s output UMI count matrix. Cells are colored by the cell clusters outputted by scReadSim (the clusters are from the real data used to train scReadSim). The synthetic cells’ deduplicated UMI counts are projected to the UMAP space defined by the same cells’ ground-truth UMI counts. We computed the Euclidean distance between each synthetic cell’s two UMAP coordinates, calculated based on the cell’s ground-truth UMI counts and deduplicated UMI counts, respectively. The mean value of the Euclidean distances of all synthetic cells is displayed for each UMI deduplication tool: a smaller value indicates that the deduplicated UMI count matrix better agrees with the ground truth UMI count matrix in UMAP visualization.

References

- [1] 10xGenomics (2019). Fresh embryonic e18 mouse brain (5k), single cell multiome atac + gene expression dataset by cell ranger arc 2.0.0. <https://www.10xgenomics.com/resources/datasets/fresh-embryonic-e-18-mouse-brain-5-k-1-standard-2-0-0>.
- [2] Cusanovich, D. A., Hill, A. J., Aghamirzaie, D., Daza, R. M., Pliner, H. A., Berletch, J. B., Filippova, G. N., Huang, X., Christiansen, L., DeWitt, W. S., et al. (2018). A single-cell atlas of in vivo mammalian chromatin accessibility. *Cell*, 174(5):1309–1324.
- [3] Davidson, N. M., Hawkins, A. D., and Oshlack, A. (2017). Supertranscripts: a data driven reference for analysis and visualisation of transcriptomes. *Genome biology*, 18(1):1–10.
- [4] Kaminow, B., Yunusov, D., and Dobin, A. (2021). Starsolo: accurate, fast and versatile mapping/quantification of single-cell and single-nucleus rna-seq data. *Biorxiv*, pages 2021–05.
- [5] Pertea, G. and Pertea, M. (2020). Gff utilities: Gffread and gffcompare. *F1000Research*, 9.
- [6] Sims, D., Iltott, N. E., Sansom, S. N., Sudbery, I. M., Johnson, J. S., Fawcett, K. A., Berlanga-Taylor, A. J., Luna-Valero, S., Ponting, C. P., and Heger, A. (2014). Cgat: computational genomics analysis toolkit. *Bioinformatics*, 30(9):1290–1291.
- [7] Smith, T., Heger, A., and Sudbery, I. (2017). UMI-tools: Modeling sequencing errors in Unique Molecular Identifiers to improve quantification accuracy. *Genome Research*, 27(3):491–499.
- [8] Srivastava, A., Malik, L., Smith, T., Sudbery, I., and Patro, R. (2019). Alevin efficiently estimates accurate gene abundances from dscrna-seq data. *Genome biology*, 20(1):1–16.
- [9] Zheng, G. X., Terry, J. M., Belgrader, P., Ryvkin, P., Bent, Z. W., Wilson, R., Ziraldo, S. B., Wheeler, T. D., McDermott, G. P., Zhu, J., et al. (2017). Massively parallel digital transcriptional profiling of single cells. *Nature communications*, 8(1):1–12.

REVIEWERS' COMMENTS

Reviewer #2 (Remarks to the Author):

The authors have addressed my remaining concerns, and the new collapsed transcriptome simulation mode is a step in the right direction. I don't have further comments.